# Distinct roles of visual, parietal, and frontal motor cortices in memory-guided sensorimotor decisions

**Michael J Goard[1,2,3,4]\*, Gerald N Pho[1,2], Jonathan Woodson[1,2], Mriganka Sur[1,2]\***

[1]Department of Brain and Cognitive Sciences, Massachusetts Institute of Technology, Cambridge, United States; [2]Picower Institute for Learning and Memory, Massachusetts Institute of Technology, Cambridge, United States; [3]Department of Molecular, Cellular, Developmental Biology, University of California, Santa Barbara, Santa Barbara, United States; [4]Department of Psychological and Brain Sciences, University of California, Santa Barbara, Santa Barbara, United States

**\*For correspondence:** michael. goard@lifesci.ucsb.edu (MJG); msur@mit.edu (MS)

**Competing interests:** The authors declare that no competing interests exist.

**Abstract** Mapping specific sensory features to future motor actions is a crucial capability of mammalian nervous systems. We investigated the role of visual (V1), posterior parietal (PPC), and frontal motor (fMC) cortices for sensorimotor mapping in mice during performance of a memory-guided visual discrimination task. Large-scale calcium imaging revealed that V1, PPC, and fMC neurons exhibited heterogeneous responses spanning all task epochs (stimulus, delay, response). Population analyses demonstrated unique encoding of stimulus identity and behavioral choice information across regions, with V1 encoding stimulus, fMC encoding choice even early in the trial, and PPC multiplexing the two variables. Optogenetic inhibition during behavior revealed that all regions were necessary during the stimulus epoch, but only fMC was required during the delay and response epochs. Stimulus identity can thus be rapidly transformed into behavioral choice, requiring V1, PPC, and fMC during the transformation period, but only fMC for maintaining the choice in memory prior to execution.

## Introduction

The ability to use sensory input to guide motor action is a principal task of the nervous system. Simple sensorimotor transformations, such as the patellar reflex, can be mediated by simple neural circuits within the peripheral nervous system. However, more sophisticated sensorimotor decisions, like using a traffic signal to guide future driving maneuvers, often requires mapping specific sensory features to motor actions at a later time, and are thought to require more complex neural circuits extending into the cerebral cortex (*Gold and Shadlen, 2007*; *Andersen and Cui, 2009*; *Romo and de Lafuente, 2013*).

Over the past several decades, a number of researchers have measured neural activity during memory-guided sensorimotor decisions using delayed-response and working memory tasks. However, despite the wealth of research in this area, there are a number of unresolved questions. First, it is unclear which regions are responsible for sensorimotor transformation. For example, single-unit electrophysiological recordings (*Shadlen and Newsome, 2001*; *Freedman and Assad, 2006*; *Gold and Shadlen, 2007*; *Andersen and Cui, 2009*; *Bennur and Gold, 2011*) and pharmacological inactivation (*Li et al., 1999*; but see *Chafee and Goldman-Rakic, 2000*) studies in non-human primates have implicated posterior parietal cortex (PPC) in the mapping sensory input to appropriate motor responses. However, recent rat auditory (*Erlich et al., 2015*) and mouse whisker (*Guo et al., 2013*) studies have challenged this view, finding no role for PPC in memory-guided sensorimotor

decisions. Rather, a number of studies suggest that cortical (*Guo et al., 2013*; *Murakami et al., 2014*; *Zagha et al., 2015*) and subcortical (*Znamenskiy and Zador, 2013*; *Kopec et al., 2015*) motor regions might be the key site for sensorimotor transformation. In contrast, PPC does appear to be necessary for visual sensorimotor decision tasks in mice (*Harvey et al., 2012*) and rats (*Raposo et al., 2014*); though the proximity of PPC to secondary visual regions makes it difficult to clearly isolate the effect of PPC inactivation (see discussion in *Erlich et al., 2015*). There are several possibilities for the discrepancies seen in the previous studies, including differences in species, sensory modality, and behavioral task. In order to further investigate the locus of sensorimotor transformation, we hoped in this study to take a more comprehensive approach toward measuring and perturbing activity across sensory, parietal association, and motor cortical regions during a delayed-response task.

A second unresolved issue lies in determining which region(s) are responsible for maintaining task-relevant information in the delay period between the stimulus and response. For decades, researchers have used delayed-response tasks to study the maintenance of information (short-term memory), and have observed the presence of sustained neural activity in distributed cortical and subcortical structures; including the prefrontal cortex (*Fuster and Alexander, 1971*; *Kojima and Goldman-Rakic, 1982*; *Funahashi et al., 1989*; *Miller et al., 1996*; *Romo et al., 1999*; *Sreenivasan et al., 2014*), parietal cortex (*Constantinidis and Steinmetz, 1996*; *Snyder et al., 1997*; *Chafee and Goldman-Rakic, 1998*; *Shadlen and Newsome, 2001*; *Harvey et al., 2012*), sensory (*Nakamura and Colby, 2000*; *Super et al., 2001*) and motor cortices (*di Pellegrino and Wise, 1993*; *de Lafuente and Romo, 2005*; *Hernandez et al., 2010*; *Guo et al., 2013*; *Erlich et al., 2015*; *Li et al., 2015*), as well as several subcortical regions (*Fuster and Alexander, 1973*; *Kawagoe et al., 1998*). Building on this work, theoretical studies have yielded biologically-plausible models describing how the sustained neural activity can be generated and maintained to support short-term memory (*Compte et al., 2000*; *Goldman et al., 2003*; *Wang, 2008*). However, recent studies in rodents have indicated that some of the regions traditionally thought to be crucial for short-term memory maintenance, such as parietal and prefrontal cortices, appear not to play a major role in some tasks (*Guo et al., 2013*; *Liu et al., 2014*; *Erlich et al., 2015*), though other studies have found an important role for sustained activity in motor regions consistent with theoretical models (*Kopec et al., 2015*; *Li et al., 2016*). Is the sustained activity observed in parietal and prefrontal cortex epiphenomenal? Or might the difference stem from another aspect of the task (e.g., modality)? We reasoned that we could help resolve this issue by measuring and perturbing activity in distributed cortical regions during a visual delayed-response task.

To leverage the genetic tractability of the mouse toward a better mechanistic understanding of memory-guided sensorimotor decisions, we modified a visual discrimination task used for head-fixed mice (*Andermann et al., 2010*) by using a retractable spout to separate the components of the task into discrete stimulus, delay, and response epochs. We took advantage of high sensitivity genetically-encoded calcium indicators (*Chen et al., 2013*) to enable high-yield 2-photon volume scanning approach (*Kampa et al., 2011*). This allowed us to image from large populations of neurons in sensory, association and frontal motor regions during discrete epochs of a memory-guided sensorimotor decision task, including stimulus encoding, delay and behavioral response.

Although measurement of neural activity is an important first step toward defining task-related regions, the presence of neural activity does not prove that a given region plays a causal role in mediating behavior. For example, it is possible that sustained activity seen in one region reflects sustained activity in a connected region without playing a direct role in the maintenance of mnemonic information. Microstimulation studies have revealed that activation of neurons in certain regions is sufficient to influence task performance (*Bisley et al., 2001*; *Hanks et al., 2006*), but does not directly address whether the region is necessary. Other studies have used surgical lesions, pharmacological inactivation or tissue cooling to reveal the cortical regions necessary for performance of short-term memory maintenance in rodents and non-human primates (*Fuster and Alexander, 1973*; *Bauer and Fuster, 1976*; *Sakurai and Sugimoto, 1985*; *Li et al., 1999*; *Chafee and Goldman-Rakic, 2000*; *Gisquet-Verrier and Delatour, 2006*; *Harvey et al., 2012*). However, while these studies have been important for establishing the anatomical framework of memory maintenance, these manipulations have limited spatial and temporal resolution. In particular, activity could not be silenced rapidly enough to test specific epochs of the task (e.g., delay period), and it is thus unclear whether a given manipulation affects stimulus perception, memory maintenance, or motor output.

Recent optical inactivation approaches have revealed that the effect of cortical inactivation on behavior is crucially dependent on timing (*Sachidhanandam et al., 2013*; *Kopec et al., 2015*; *Li et al., 2016*) and whether inactivation is unilateral or bilateral (*Li et al., 2016*). To continue this approach, we used an optogenetic approach for reversibly silencing activity bilaterally in defined cortical regions with precise temporal control (*Zhao et al., 2011*). Using inactivation of bilateral cortical regions exhibiting task-related responses, we were able to determine the necessity of sensory, association, and frontal motor cortical regions during each epoch (stimulus, delay, response) of a memory-guided task.

## Results

### Calcium imaging during a memory-guided sensorimotor decision task

We trained head-fixed mice to perform a visual discrimination task with a memory-guided response (*Figure 1A*). In this task, water-restricted mice were presented a 2 s drifting grating stimulus at one of two orientations (0°, 90° from vertical), followed by a variable delay period (0-, 3-, or 6-s), at which point a lick spout was moved rapidly into reach with a linear actuator for 1.5 s (*Figure 1B*, bottom). Licking on 'go' trials (horizontal grating drifting toward 0° from vertical) was rewarded with 5–8 µl water (hit), while licking on 'no-go' trials (vertical grating drifting toward 90° from vertical) was punished with 2 µl water containing 5 mM quinine hydrochloride (false alarm; *Figure 1B*, top; *Video 1*). This structure allowed the separation of each trial into 'stimulus', "delay", and 'response' epochs. Notice that the stimulus-choice association is fixed, so correct performance does not require memory of the stimulus during the delay period (memory of the planned response is sufficient). After extensive training (117 ± 11 behavioral sessions, 299 ± 25 trials per session; mean ± s.e.m.), mice reliably exhibited strong differences in licking between go and no-go trials (*Figure 1C*, top) for all delay period durations. We applied an *a priori* exclusion criteria that any mice licking continuously throughout the delay period on target trials would be excluded from the study, since this strategy would possibly obviate the short-term memory component of the task. Video analysis showed that mice did not exhibit postural changes or increased movement during the delay period (*Figure 1— figure supplement 1*; see *Video 1* for representative mouse performance). Mice did exhibit a bias toward licking (as observed previously with go/no-go tasks; *O'Connor et al., 2010*; *Huber et al., 2012*), so we quantified their performance using d-prime rather than percent correct to account for motivation and criterion (*Carandini and Churchland, 2013*) (*Figure 1C*, bottom). Performance decreased slightly with longer delays, but was well above chance for all delay durations (0 s Delay, $p<10^{-9}$; 3 s Delay, $p<10^{-9}$; 6 s Delay, $p<10^{-9}$; t-test, $n = 8$ mice across 80 sessions).

We focused our experiments on three cortical regions we expected to be important for performance of this task: (1) the primary visual cortex (V1), which is known to be important for orientation discrimination (*Glickfeld et al., 2013*); (2) the posterior parietal cortex (PPC), which receives extensive input from visual regions (*Harvey et al., 2012*; *Oh et al., 2014*; *Pho et al., 2015*), projects to motor regions (*Wang et al., 2012*), and has been implicated in sensorimotor decision tasks (*McNaughton et al., 1994*; *Shadlen and Newsome, 2001*; *Nitz, 2006*; *Gold and Shadlen, 2007*; *Whitlock et al., 2008*; *Andersen and Cui, 2009*; *Raposo et al., 2014*; *Hanks et al., 2015*) and (3) the frontal motor cortices (fMC), which include regions known to be crucial for voluntary licking behaviors (*Komiyama et al., 2010*; *Guo et al., 2013*).

We identified parietal cortex on the basis of stereotaxic coordinates from previous studies (*Harvey et al., 2012*). Note that this region has weak visual responses and has also been classified as a secondary visual region (AM) (*Wang et al., 2012*; *Garrett et al., 2014*). However, in addition to visual inputs, retrograde tracing from our lab (unpublished results) and others (*Harvey et al., 2012*) has revealed that the region receives input from auditory, somatosensory, secondary motor, and frontal cortices, as well as the lateral posterior thalamic nucleus. Since the parcellation of rodent frontal and motor cortices is a subject of debate in the field (*Brecht, 2011*), and both medial and lateral regions have been implicated in licking (*Komiyama et al., 2010*) we define fMC on the basis of stereotaxic coordinates (including primary and secondary motor regions) and remain agnostic as to its precise homology with primate frontal and motor cortices. Nonetheless, several studies have indicated that rodent fMC plays an important role in tasks involving perceptual decisions and memory (*Kepecs et al., 2008*; *Guo et al., 2013*; *Erlich et al., 2015*; *Li et al., 2015*).

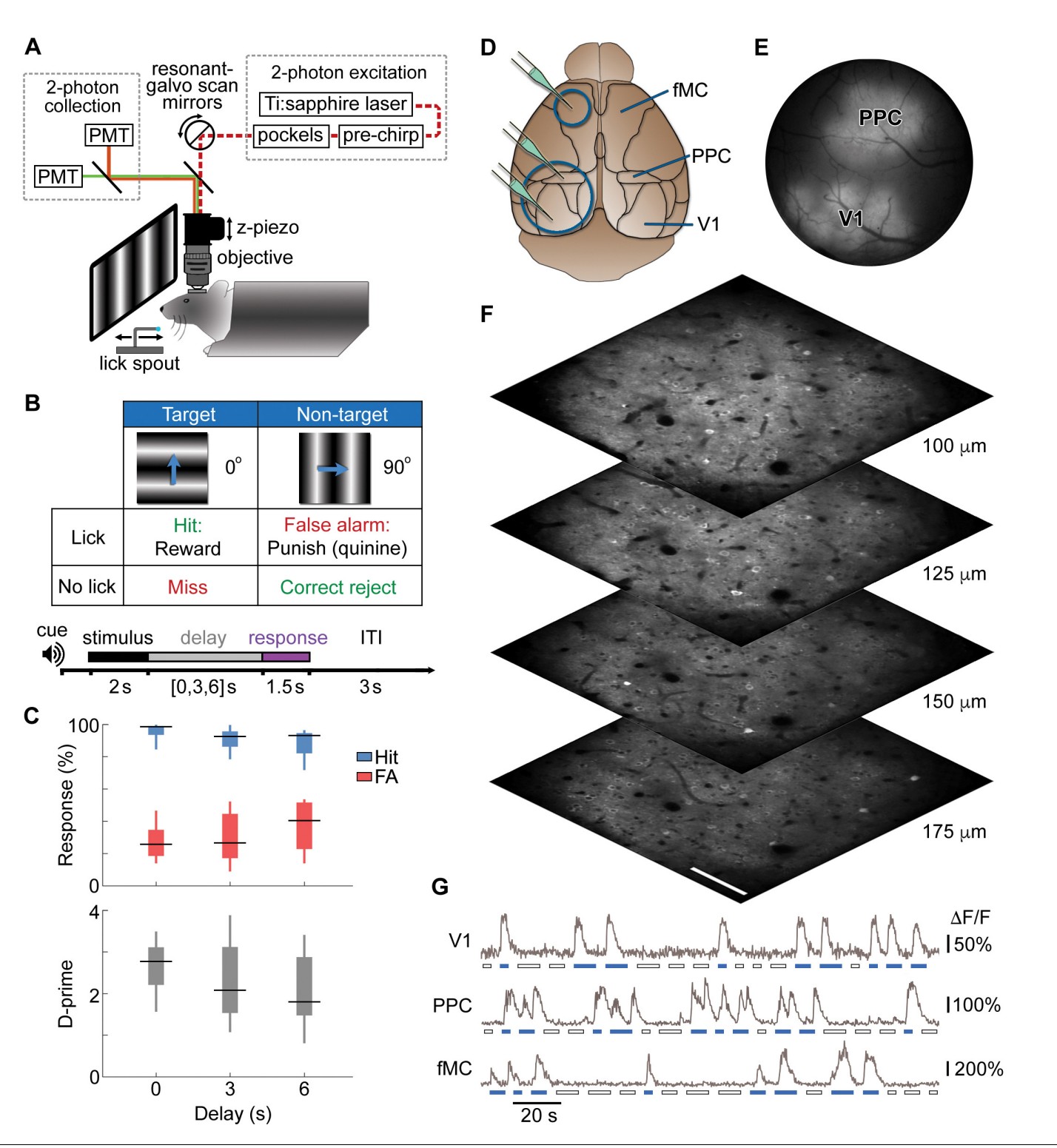

**Figure 1.** Calcium imaging during a memory-guided task. (**A**) Experimental setup for 2-photon imaging in head-fixed mice performing a memory-guided visual discrimination task. The mode-locked infrared laser (dotted red line) was raster scanned at a high rate using resonant-galvo scan mirrors, which were synchronized to a z-piezo to allow volumetric imaging. A retractable lick spout was used to restrict the timing of behavioral responses to a specific epoch of the task. (**B**) Contingency table (top) and trial structure of the memory-guided visual discrimination task (bottom). Licks to the target orientation were rewarded with water, whereas licks to the non-target orientation were punished with quinine. Trials consisted of stimulus, delay, and response epochs, and the retractable spout was within reach only during the response epoch. Trials of three different delays (0, 3, or 6 s) were randomly

*Figure 1 continued on next page*

*Figure 1 continued*

interleaved. (C) Average behavioral performance (n = 8 mice across 80 sessions). Top, Response rate for target stimuli (hit rate; blue) and non-target stimuli (false alarm rate; red). Bottom, D-prime for delays of 0–6 s; performance is significantly above chance for all delays ($p < 10^{-9}$, *t*-test). (D) Location of cranial windows (blue circles) and AAV-GCaMP6s injections (green pipettes) in primary visual cortex (V1), posterior parietal cortex (PPC), and frontal motor cortex (fMC). Schematic modified from Allen Brain Atlas (http://www.brain-map.org). (E) Example wide-field epifluorescence image of GCaMP6s expression in both V1 and PPC (window diameter, 4 mm). (F) Four imaging planes (25 μm apart) within V1 acquired at a stack rate of 5 Hz using a resonance scanner synchronized with a Z-piezo. Images were collected at 2x zoom for clarity. Scale bar, 100 μm. (G) Sample raw ΔF/F traces from V1, PPC, and fMC during interleaved target (blue bars) and non-target (white bars) trials of varying delay duration.

The following figure supplements are available for figure 1:

**Figure supplement 1.** Video analysis of movement during delay period.

**Figure supplement 2.** Large-scale volume imaging of neural activity.

**Figure supplement 3.** Weak calcium responses in identified cortical interneurons during volume scanning.

To measure neural activity in layer 2/3 of cortex during task performance, a craniotomy was made over one or more of regions V1, PPC and fMC (*Figure 1D*) after completion of training. Stereotaxically-guided microinjections of adeno-associated virus (AAV) containing the genetically-encoded calcium indicator GCaMP6s (*Chen et al., 2013*) were made in V1 (n = 5 mice), PPC (n = 6) and/or fMC (n = 4), and sealed cranial windows were made over V1 and PPC (*Figure 1E*) or fMC. To increase the number of recorded neurons, we used a volume scanning approach (*Kampa et al., 2011*) that allowed us to image four imaging planes (separated by 20–25 μm) to simultaneously sample hundreds of GCaMP6s-infected neurons within an 850 μm x 850 μm x 60–75 μm volume at a sample rate of 5 Hz (*Figure 1F*; *Figure 1—figure supplement 2*; *Video 2*). Images were corrected for X-Y movement and regions of interest were assigned to cell somata based on a pixel-wise activity map calculation, yielding fluorescence traces from individual neurons that were active during the imaging session (26 sessions, active neurons per session: 352 ± 40, mean ± s.e.m.). A total of 9,150 active neurons were imaged in regions V1 (n = 2,695 neurons), PPC (n = 3,552), and fMC (n = 2,903), of which 3,049 (33.3%) exhibited trial-locked responses significantly different from baseline (see Materials and methods for inclusion criteria; *Figure 1G*). Pilot experiments in transgenic mice expressing tdTomato in PV+ and SOM+ interneurons revealed that interneuron calcium signals measured with volume scanning were considerably smaller than tdTomato-negative (putatively excitatory) neurons (*Figure 1—figure supplement 3*, though see *Peron et al., 2015*), and therefore the vast majority of task-responsive cells were likely excitatory pyramidal neurons.

## Single neurons exhibit heterogeneous trial-evoked responses

To investigate how regions V1, PPC, and fMC encode task-relevant variables, we analyzed the activity of single neurons in each region during task performance (*Figure 2*). The majority (63%, *Figure 3*, see below for description of classification procedure) of V1 neurons with increased task-evoked activity exhibited robust and reliable responses during the stimulus epoch but not during other epochs of the task, with similar responses across 0, 3, and 6 s delays (*Figure 2A,B*). However, there was also a sizable fraction (37%) of neurons that exhibited activity during the other task epochs, particularly during the response epoch (*Figure 2C*). Interestingly, there was an additional fraction (53% of responsive neurons) of neurons that were suppressed throughout the stimulus and delay periods in a delay duration-dependent fashion (*Figure 2D*). To our knowledge, neurons exhibiting delay-dependent suppression have not been previously described in V1 despite their prevalence in our sample. To ensure that the suppressed activity pattern was not an artifact of the calcium imaging technique, we carried out single-unit electrophysiological recordings in two mice using silicon probes in V1 during behavior and found a sizable fraction of neurons with suppressed responses during behavior (10/21 single units; see *Figure 2—figure supplement 1*).

Neurons in PPC were even more heterogeneous, with a large number (48% of enhanced neurons) of neurons exhibiting activity during both stimulus and response epochs (*Figure 2E,F*), as well as some neurons (11% of enhanced neurons) exhibiting delay-dependent enhanced activity

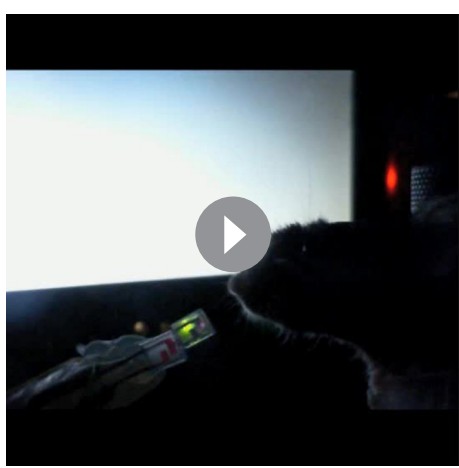

**Video 1.** Example video of a mouse performing the memory-guided visual discrimination task. Task epoch (stimulus, delay, choice), stimulus identity (target, non-target), and trial types (hit, correct reject) are annotated.

(*Figure 2G*), as has previously been observed in primate experiments in delayed-response tasks (*Chafee and Goldman-Rakic, 1998*; *Constantinidis and Steinmetz, 1996*). Finally, we again saw neurons that exhibited suppressed delay-dependent activity during the task (*Figure 2H*), though they were much less prevalent compared to V1 (9% vs. 53% for PPC and V1, respectively).

Neurons in fMC likewise exhibited heterogeneous responses during multiple task epochs. As expected in motor regions, a plurality (47% of enhanced neurons) of neurons responded solely during the response epoch (*Figure 2I*). However, a substantial fraction (33% of enhanced neurons) of neurons responded not only during the response epoch, but also in a sustained manner throughout the delay period (*Figure 2J*), including neurons that exhibited sustained activity during the delay period of the task, and actually showed decreased activity during the response period (*Figure 2K*). Note that the sustained activity cannot be attributed to the slow decay of the GCaMP6 indicator after the initial stimulus response, as the duration of the sustained activity varies parametrically with the duration of the delay period. Sustained activity was also observed using a calcium indicator with a faster time course (GCaMP6f, data not shown). Finally, we again observed a subset (39% of responsive neurons) of neurons with suppressed delay-dependent activity in fMC (*Figure 2L*).

To characterize the diversity of trial-evoked responses and their relative prevalence in each region, we clustered the neurons by their response characteristics. We pooled the significant responses (enhanced and suppressed) from all three areas, and then used PCA-based denoising and hierarchical clustering to delineate distinct response types (see Materials and methods). Hierarchical clustering revealed six distinct response types, four of them exhibiting enhanced activity and two exhibiting suppressed activity (*Figure 3A*). To reveal the archetypical response of each group, we averaged the normalized responses of all neurons within each cluster, revealing four enhanced response classes: stimulus-driven (*Stim only*), response-driven (*Resp only*), stimulus- and response-driven (*Stim+Resp*), and delay-driven (*Delay*), as well as two suppressed response classes: delay-sensitive stimulus-driven (*Supp. Delay early*) and delay-sensitive response-driven (*Supp. Delay late*; *Figure 3B*). These clusters have strong face validity, as the average responses (*Figure 3B*) are qualitatively similar to observed single-neuron responses (*Figure 2*; *Figure 3—figure supplement 1–3*). Although neurons that were active during the response epoch were predominantly selective for 'go' trials, there were response-driven neurons that were selective for 'no-go' trials as well

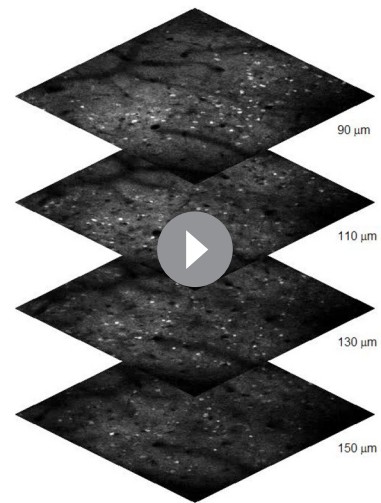

**Video 2.** Example video of volume imaging of GCaMP6s-labelled neurons in V1 (spontaneous activity). Each frame is 850 × 850 μm, with 20 μm separation between frames in the Z-dimension. Display speed is 10x real time.

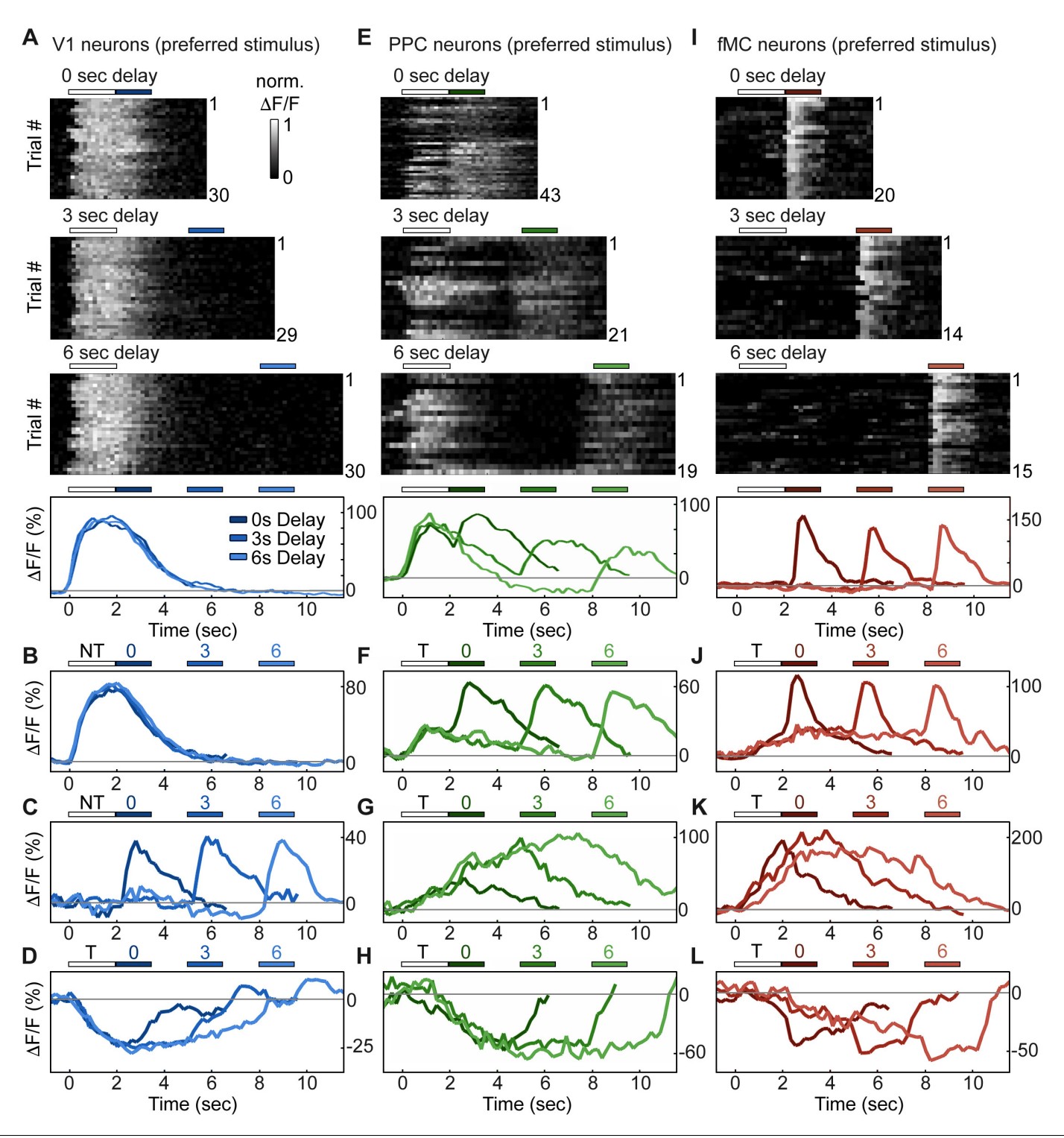

**Figure 2.** Single neurons exhibit heterogeneous trial-evoked responses. (**A**) Trial-to-trial and average responses from a single V1 neuron. Top three plots, stacked single-trial ΔF/F responses to the preferred stimulus (correct trials only), grouped into 0-, 3-, and 6-second delay trials. The neuron is active during the stimulus for all three delays. Colored bars above plots indicate time of visual stimulus (white) and spout extension (blue, shade indicates delay period). Bottom plot, overlay of mean ΔF/F responses during 0-, 3-, and 6-s delay trials. Shade of average traces indicates delay period (dark blue, 0-s delay; medium blue, 3-s delay, light blue, 6-s delay). Additional V1 example neurons exhibiting activity during the stimulus period (**B**), response period (**C**), or suppressed activity throughout the delay period (**D**). Color shade indicates delay as in (**A**). (**E**) Same as (**A**) but for a PPC neuron.

*Figure 2 continued on next page*

*Figure 2 continued*

This neuron is active during both stimulus and response period. Colored bars above plots indicate time of visual stimulus (white) and spout extension (green, shade indicates delay period). Bottom plot, overlay of mean ΔF/F responses during 0-, 3-, and 6-s delay trials. Shade of average traces indicates delay period (dark green, 0-s delay; medium green, 3-s delay, light green, 6-s delay). Additional PPC example neurons exhibiting activity during the stimulus and response period (F), sustained activity during the delay period (G), or suppressed activity throughout the delay period (H). Color shade indicates delay as in (E). (I) Same as (A) but for a fMC neuron. This neuron is active during the response period. Colored bars above plots indicate time of visual stimulus (white) and spout extension (red, shade indicates delay period). Bottom plot, overlay of mean ΔF/F responses during 0-, 3-, and 6-s delay trials. Shade of average traces indicates delay period (dark red, 0-s delay; medium red, 3-s delay, light red, 6-s delay). Additional fMC example neurons exhibiting activity during the delay and response period (J), during only the delay period (K), or suppressed activity throughout the delay period (L). Color shade indicates delay as in (I).

The following figure supplement is available for figure 2:

**Figure supplement 1.** Electrophysiological recordings of enhanced and suppressed neurons in V1.

(*Figure 3—figure supplement 1–3*, plots with preferred responses to non-target, 'N', stimulus), suggesting that the response-driven activity is not entirely related to motor activity. There were clear differences in both enhanced and suppressed response types between regions, such as V1 containing a large number of '*Stim only*' and '*Supp. Delay early*' neurons, while fMC had greater numbers of '*Delay*', '*Resp only*', and '*Supp. Delay late*' neurons (*Figure 3C*). However, there was a surprising amount of heterogeneity as well, with all three regions containing a fraction of almost all response types.

## Suppressed neurons exhibit non-selective responses

We next investigated what role neurons exhibiting suppressed activity (975/3049 neurons, 32.0%; *Figure 2D,H,L*) play in encoding task-relevant information. We found that the enhanced and suppressed neurons across regions exhibited striking differences in selectivity. The vast majority of enhanced neurons showed a strong preference for a particular trial type (hit vs. correct reject, only correct trials included in analysis), with little or no response to the non-preferred trial type (*Figure 4A*). However, the majority of suppressed neurons showed very similar responses to both trial types, with little difference between 'preferred' and 'non-preferred' trial types (*Figure 4B*). We used a selectivity index ranging from 0 (responds equally to preferred and non-preferred stimuli) to 1 (only responds to preferred stimuli) to quantify the selectivity of each neuron (see Materials and methods), revealing that enhanced neurons were significantly more selective than suppressed neurons in all three regions (*Figure 4C*; V1: Enh. median SI = 0.67, Supp. median SI = 0.16, $p<10^{-9}$; PPC: Enh. median SI = 0.91, Supp. median SI = 0.31, $p<10^{-9}$; fMC: Enh. median SI = 0.99, Supp. median SI = 0.53, $p<10^{-9}$; Wilcoxon rank-sum test).

One possibility for the presence of the suppressed responses is that local inhibition is being recruited by the increased activity of the enhanced neurons during task performance. If this were the case, we would expect that the time course of the suppressed responses would closely follow that of the enhanced responses. We estimated the response latency of the suppressed neurons, and found that the latencies were slow (>400 ms) and uncorrelated to the enhanced population latency in the cortical region (*Figure 4D*). We also found that suppressed neurons exhibited a suppression throughout the delay period (in a duration-dependent manner; *Figure 2D,H,L*), even when this pattern was not present in the enhanced neurons of the same region (*Figure 4E*). Taken together, these findings suggest that the suppressed responses are not simply reflecting inhibition from local excitatory responses, but rather are the result of more complex dynamics; possibly low-latency, delay-dependent inputs from distal regions.

## Enhanced neurons show regional differences despite local heterogeneity

The lack of trial type selectivity observed in suppressed neurons indicates that there will be little information about stimulus identity or future motor response in the activity of these neurons. However, the enhanced neurons are highly selective, and likely represent these task-relevant variables in a more robust manner. To further investigate the role of enhanced neurons in different cortical

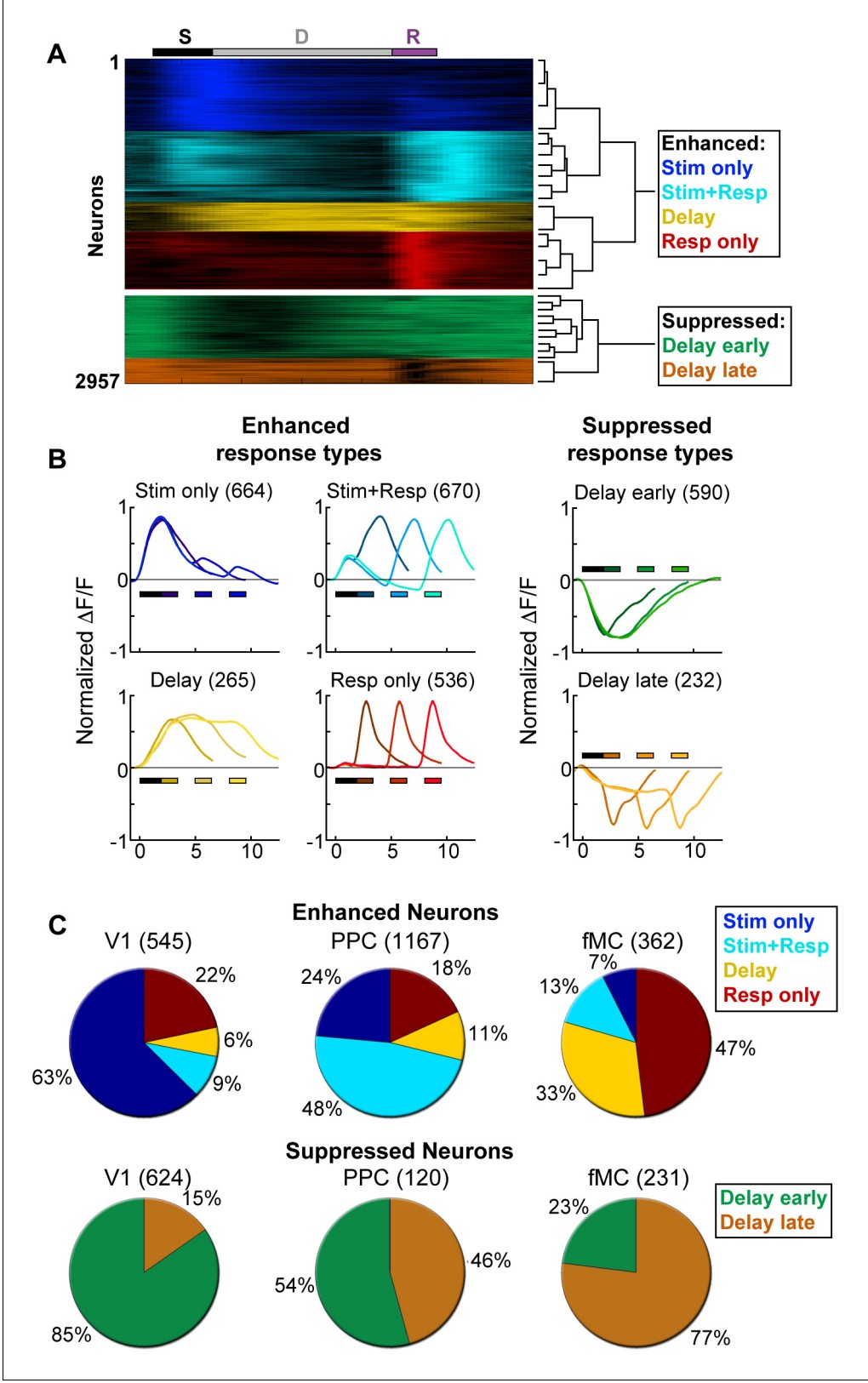

**Figure 3.** Clustering reveals distinct response types distributed between regions. Hierarchical clustering was performed on the first 20 principal components (explaining >98% of the variance), revealing six separable clusters. Further cluster division resulted in separation of very similar response types. (**A**) Normalized calcium response to

*Figure 3 continued on next page*

*Figure 3 continued*

the preferred stimulus (correct trials only) of each neuron for the 6 s delay condition, separated by cluster identity (color indicates cluster). Bars at top of plot indicate time of visual stimulus (white), delay (grey), and spout extension (purple). Right inset shows a dendrogram resulting from the hierarchical clustering procedure and cluster names. (B) Average normalized response of each cluster for all delay durations, separated by enhanced and suppressed clusters. Titles indicate the name of the response type cluster and the number of neurons included. Response color indicates cluster identity. Colored bars indicate time of visual stimulus (white) and spout extension (color indicates cluster, shade indicates delay duration). (C) Proportion of neurons in V1 (*n* = 1169), PPC (*n* = 1287) and fMC (*n* = 593) of each response class.

The following figure supplements are available for figure 3:

**Figure supplement 1.** Example V1 neuron responses.
**Figure supplement 2.** Example PPC neuron responses.
**Figure supplement 3.** Example fMC neuron responses.

regions, we investigated the population responses within regions by pooling the neurons across imaging sessions from different animals.

To investigate the encoding of task variables at the population level, we averaged and normalized the preferred responses of enhanced task-driven neurons across all correct trials of equivalent delay (hit or correct reject, depending on preferred response) and sorted them first by cortical region, and then by latency to peak response (*Figure 5A*; suppressed neuron responses shown in *Figure 5—figure supplement 1*). While V1 neurons preferred go and no-go trials in roughly equal numbers, we found that PPC and fMC neurons were highly biased toward the go trials (*Figure 5B*). Since the target and non-target stimuli have similar visual saliency, this suggests that these regions may encode task-related variables other than stimulus identity. Indeed, although the responses within regions were heterogeneous (*Figure 3*), there were clear differences in the average responses between regions (*Figure 5C*). Specifically, V1 neurons were predominantly active during the stimulus epoch, while PPC and fMC exhibited more heterogeneous responses spanning stimulus, delay, and response epochs. Although sustained delay-period activity could be observed across all cortical regions, it was most prevalent in fMC, both in single-unit (*Figure 3C*) and population (*Figure 5C*) activity. Finally, there were considerable differences in latency to significant response, with the first mode of the latency distribution increasing from V1 to PPC to fMC for all delay durations (V1: 89 ± 10 ms, PPC: 244 ± 15 ms, fMC: 827 ± 177 ms, mean across delays; regions all significantly different, V1 x PPC: $p<10^{-9}$, V1 x fMC: $p<10^{-9}$, PPC x fMC: $p<10^{-9}$, Wilcoxon rank-sum test; *Figure 5D*).

## Population error-trial analysis reveals distinct encoding dynamics in each region

To further investigate how neural activity might reflect the encoding of the stimulus identity and behavioral choice (including planning of the response before it is made), we analyzed the modulation of responses in all regions during error trials. We hypothesized that if neurons were simply encoding stimulus identity, there would be no difference in activity during correct trials and error trials containing the same visual stimulus (i.e., hit vs. miss; correct reject vs. false alarm). Indeed, V1 neurons exhibit similar responses to correct and error trials with the same stimulus, while neurons in PPC and fMC show very different responses on the two trial types (*Figure 6A*, left and middle panels corresponding to miss vs. hit trials; Pearson correlation between miss and hit trials, V1: 0.47 ± 0.03, PPC: 0.05 ± 0.01, fMC: 0.06 ± 0.02). Similarly, we hypothesized that if neurons encoded the animal's behavioral choice independently of the stimulus shown, we would expect there to be little difference between correct trials and error trials with the same motor response (i.e., hit vs. false alarm; correct reject vs. miss). Neuronal responses in PPC and fMC appear to exhibit more similarity than V1 for trial types with the same motor response (*Figure 6A*, middle and right panels corresponding to hit vs. false alarm trials; Pearson correlation between hit and false alarm trials, V1: 0.23 ± 0.03, PPC: 0.47 ± 0.01, fMC: 0.46 ± 0.02).

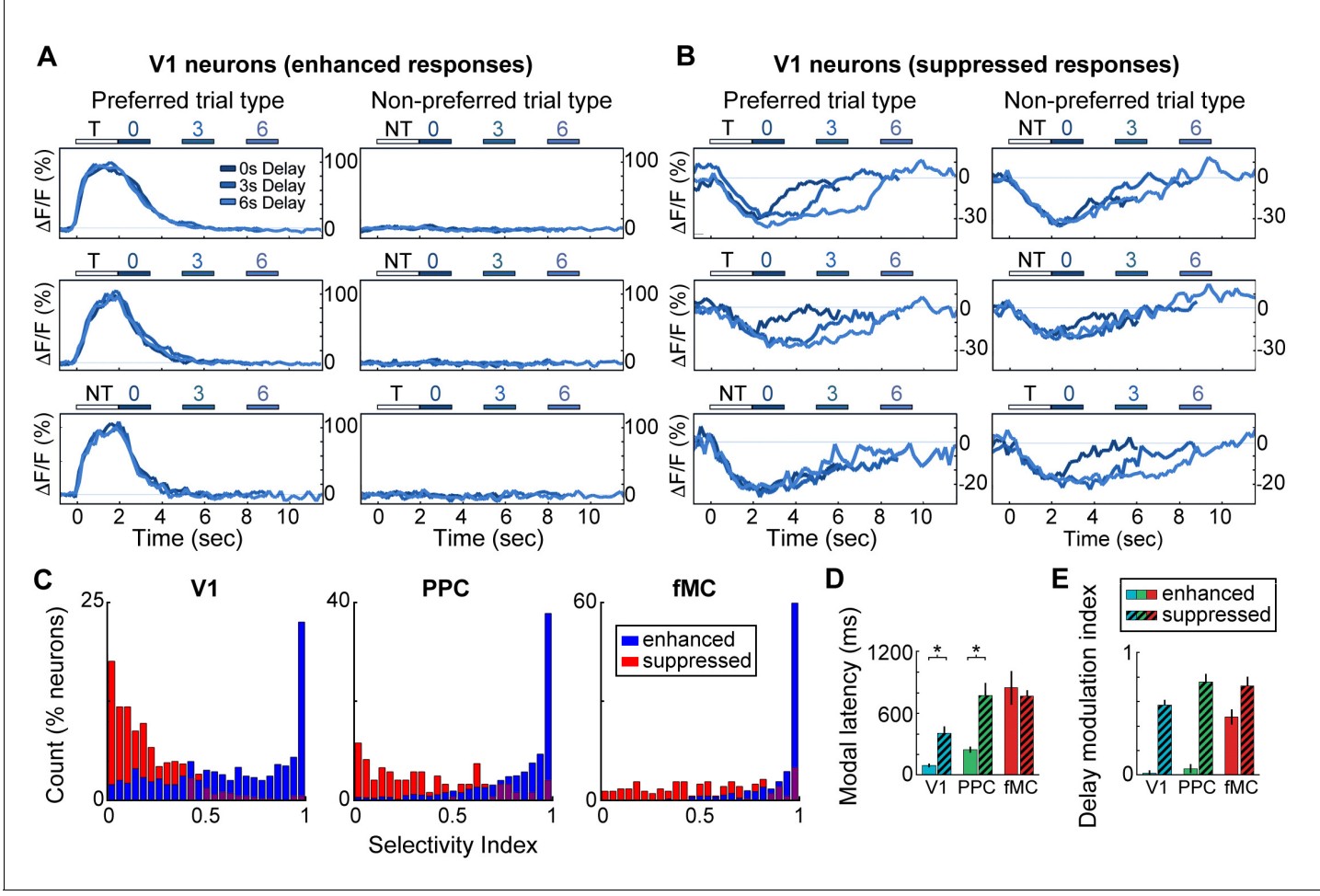

**Figure 4.** Neurons with suppressed activity are much less selective than neurons with enhanced activity. (**A**) Three example V1 neurons (top, middle, bottom) with enhanced task-driven activity. Responses shown for preferred stimulus (left) and non-preferred stimulus (right), including only correct trials. Each plot shows an overlay of the mean ΔF/F responses during 0-, 3-, and 6-s delay trials. Colored bars above plots indicate time of visual stimulus (white; T, target stimulus; NT, non-target stimulus) and spout extension (blue, shade indicates delay period). Shade of average traces indicates delay period (dark blue, 0-s delay; medium blue, 3-s delay; light blue, 6-s delay). (**B**) Three example V1 neurons (top, middle, bottom) with suppressed task-driven activity. Responses shown for preferred stimulus (left) and non-preferred stimulus (right), including only correct trials. Color shade indicates delay as in (**A**). (**C**) Histograms of selectivity index for enhanced (blue) and suppressed (red) neurons for each of the three regions (V1, left; PPC, middle; fMC, right). (**D**) Modal latency for enhanced (solid) and suppressed (hashed) neurons for each of the three regions (V1, blue; PPC, green; fMC, red). Latencies for enhanced and suppressed populations differ in V1 and PPC, but not fMC. (**E**) Delay modulation index for enhanced (solid) and suppressed (hashed) neurons for each of the three regions (V1, blue; PPC, green; fMC, red). The delay modulation increases for enhanced neurons from V1 to PPC to fMC, but is high in all three regions for suppressed neurons. Wilcoxon signed-rank test with Bonferroni correction used for all statistical tests. For bar plots, bars indicate mean ± bootstrap-estimated S.E.M.

To quantify the encoding of task variables as a function of time in each region, we used an ideal observer analysis to determine how well we could decode the stimulus identity independent of the behavioral choice (*Figure 6B*) and behavioral choice independent of stimulus identity (*Figure 6C*) using only the responses from randomly selected populations of neurons (ranging from population size of 1 to 100, see Materials and methods for details of decoding procedure). In V1, stimulus identity could be decoded with perfect accuracy given a sufficient number of neurons (*Figure 6B*, left), while behavioral choice was only weakly encoded (*Figure 6C*, left). The stimulus identity encoding peaked during the visual stimulus and then gradually declined throughout the trial (statistically significant from baseline from 0.2–9.2 s, mean value greater than 95% CI of shuffled permutations for consecutive time points, see Materials and methods), likely due to the slow decay of the GCaMP6s indicator, while the behavioral choice encoding gradually increased, peaking during the response

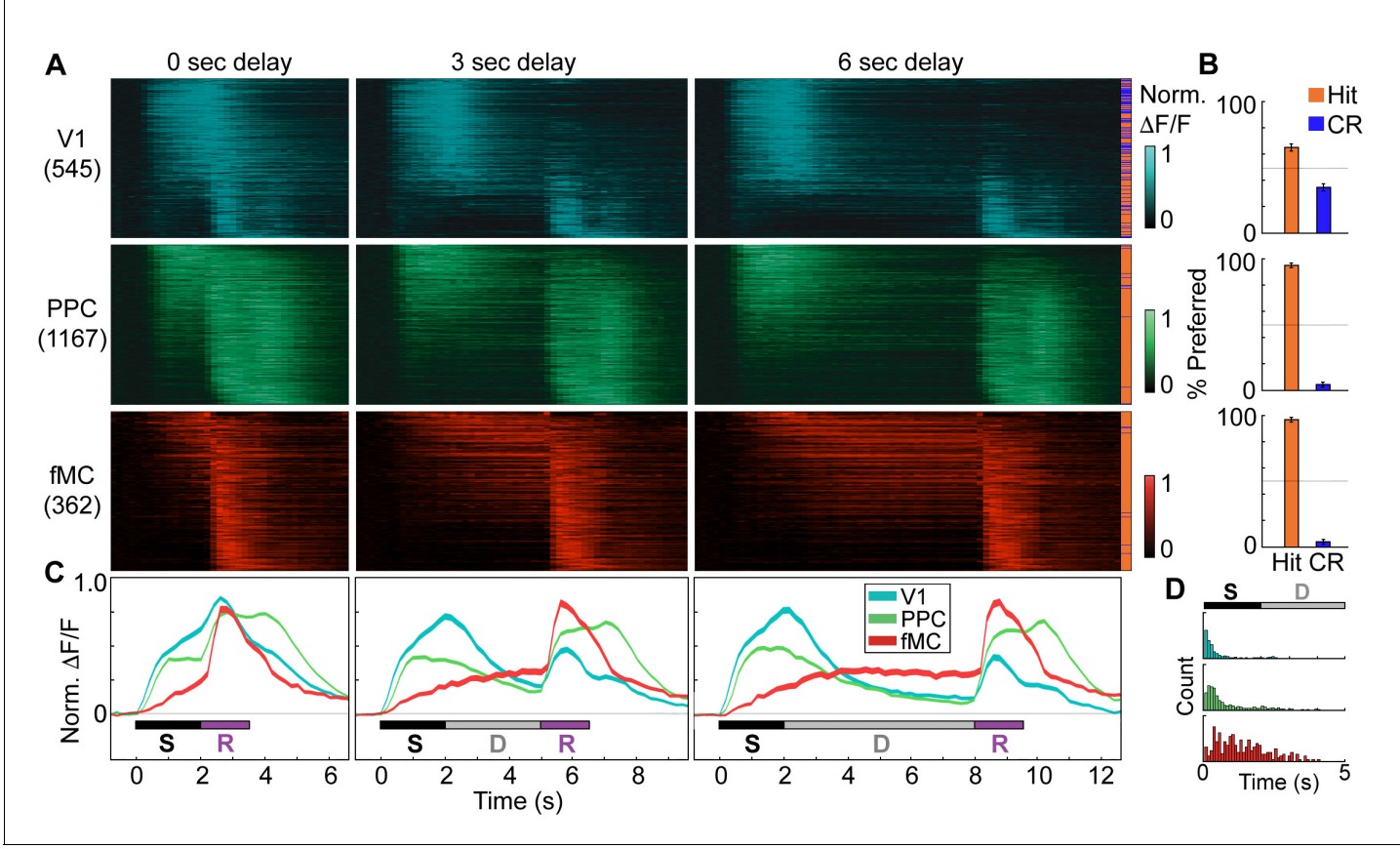

**Figure 5.** Distinct population dynamics in regions V1, PPC, and fMC. (A) Normalized preferred calcium responses of all significantly responsive neurons showing enhanced responses (pooled across all 8 mice) in V1 (top, *n* = 545), PPC (middle, *n* = 1167) and fMC (bottom, *n* = 362), across all three delay durations (0-, 3-, 6-s; left, middle, and right, respectively). Only correct trials were included. For each neuron, traces were normalized to the peak of each cell's trial-averaged response (colorbar on right inset). For each area, neurons were sorted by the time of the peak response. Sidebar indicates preferred trial for each neuron (Hit, orange; CR, blue). (B) Proportion of task-responsive neurons in each brain region that preferred Hit (orange) vs. Correct reject (CR, blue) trials. There was a strong bias for neurons in PPC and fMC to toward hit trials. (C) Mean population response of each brain region (V1, blue; PPC, green; fMC; red), across the three delay durations (line boundaries indicate mean ± bootstrap-estimated s.e.m.). Colored bars indicate the times of visual stimulus (white), delay (gray), and spout extension (purple). (D) Histogram of latency to the significant response of each brain region (only neurons with significant responses during stimulus or delay epochs considered). The population of single neuron latencies in V1 significantly precedes PPC, which in turn precedes fMC, for all delays (*p* < 10$^{-9}$, all comparisons, Wilcoxon rank sum test). Color indicates region (V1, blue; PPC, green; fMC; red).

The following figure supplement is available for figure 5:

**Figure supplement 1.** Suppressed population responses in regions V1, PPC, and fMC.

period (statistically significant from 4.8 to 6.8 s and from 8.4 to 10.0 s; *Figure 6D*, left). Note that in all three regions, the increase in stimulus identity encoding during the response epoch is likely the byproduct of reward or punishment (which is directly associated with the stimulus identity) influencing motor activity. In PPC, both stimulus identity and behavioral choice could be decoded with moderate success (*Figure 6B,C*, middle), although the dynamics were noticeably different. Specifically, while the stimulus identity was predominantly encoded early in the trial (statistically significant from 0.6–3.4 s and 7.2–10.0 s), the behavioral choice encoding slightly lagged the stimulus identity encoding, and remained high throughout the trial, peaking during the early part of the licking response (statistically significant from 1.4–10.0 s; *Figure 6D*, middle). Note that the stimulus encoding is much weaker in PPC than in V1, likely due to the large number of neurons that jointly-encode stimulus and choice information (*Park et al., 2014*; *Raposo et al., 2014*; *Pho et al., 2015*). Finally, in fMC the stimulus identity could not be decoded above chance for the majority of the trial

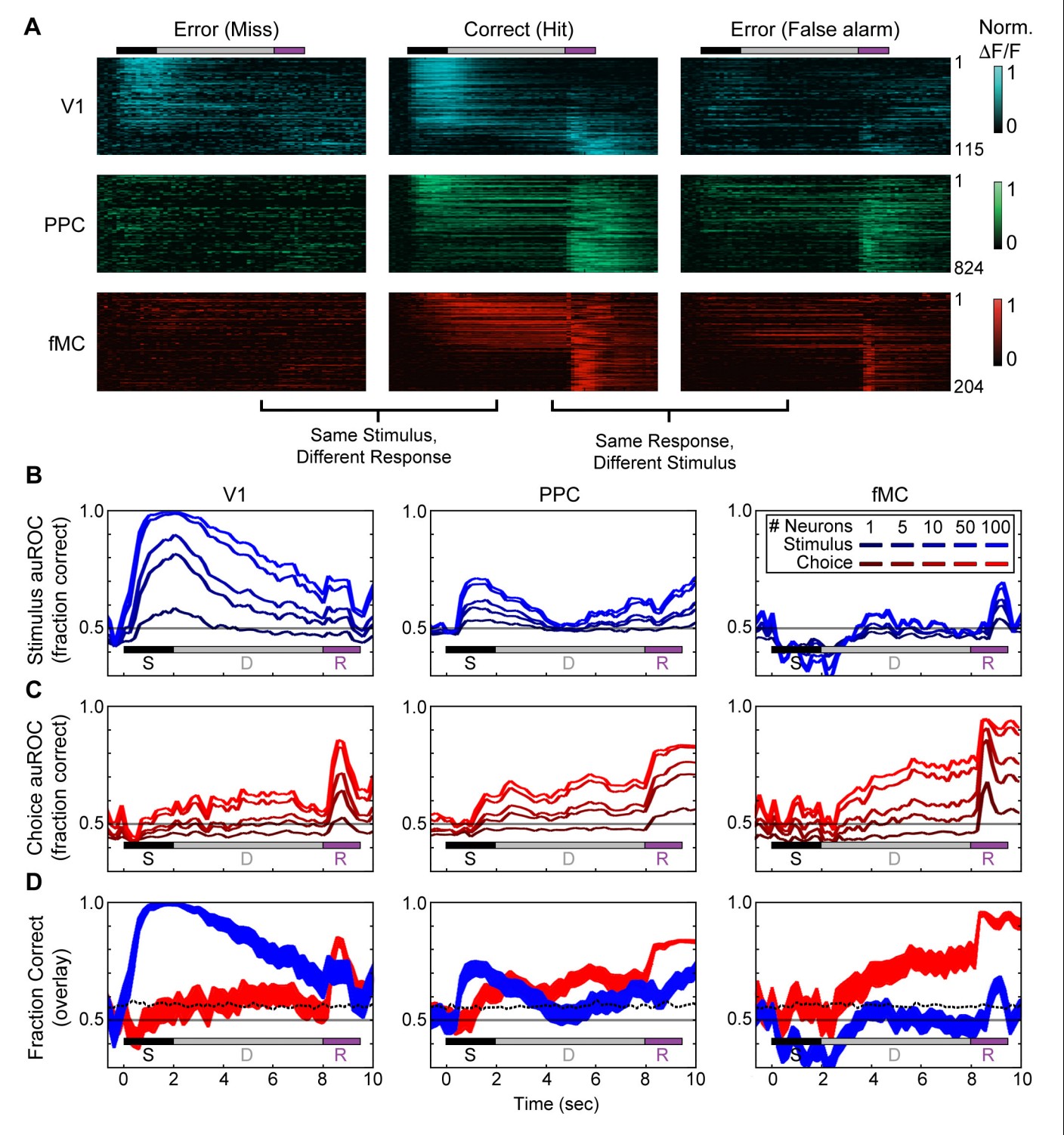

**Figure 6.** Error responses and population coding of task variables. (**A**) Responses of all target-preferring neurons (6-s delay trials only) in all three regions (V1: top, blue; PPC: middle, green; fMC: bottom, red) to correct target trials (Hit, middle), error trials with the same stimulus but different response (Miss, left), and error trials with the same response but different stimulus (False alarm, right). Color bars indicate normalized ΔF/F (0 to 1). (**B**) Ideal observer performance (area under the receiver operator characteristic, auROC) in estimating the stimulus identity (target, non-target) based only on the neural activity of significantly enhanced neurons in V1 (left), PPC (middle), and fMC (right), including randomly sampled correct and error trials). Line shading indicates neuron population size used for the decoding (dark to light blue: n = 1 to 100 neurons; see legend). (**C**) Ideal observer performance in estimating the behavioral choice (lick, no-lick) based only on the neural activity of significantly enhanced neurons in V1 (left), PPC

*Figure 6 continued on next page*

*Figure 6 continued*

(middle), and fMC (right), including randomly sampled correct and error trials). Line shading indicates neuron population size used for the decoding (dark to light red: *n* = 1 to 100 neurons). (D) Re-plot of stimulus identity and behavioral choice decoding performance for *n* = 100 neurons, with line boundaries indicating mean ± bootstrap-estimated S.E.M. Dotted lines indicate upper 95% confidence interval for 1000 permutations with shuffled labels.

(*Figure 6B*, right), while the behavioral choice could be decoded with greater accuracy in fMC than in other regions (*Figure 6C*, right). Thus, while stimulus identity decoding was only above chance during the response epoch (statistically significant from 8.8–9.4 s, see caveat on stimulus identity encoding during response epoch above), the behavioral choice could be decoded throughout the delay and response epochs (statistically significant from 3.0–10.0 s; *Figure 6D*, right).

Although calculating the auROC is an easily interpretable measure of information encoding, it is also a noisy measure and may be obscuring dynamics at a finer timescale. As a second approach to investigating the encoding dynamics of neural populations in different regions, we used targeted dimensionality reduction to re-map the high dimensional neural space into a lower dimensional subspace in which the axes correspond to task-relevant variables (*Mante et al., 2013*). This was accomplished by first using principal component analysis to reduce the high-dimensional neural space in each region to a lower-dimensional 'de-noised' subspace, then using linear regression to define orthogonal axes for 'stimulus' and 'choice' from the principal components (see Materials and methods). By measuring the difference in 'hit' and 'correct reject' trajectories along the 'stimulus' and 'choice' axes, we found once again that V1 principally encodes the stimulus identity (*Figure 7A,D*), PPC encodes a mixture of stimulus identity and behavioral choice (*Figure 7B,E*), and fMC predominantly encodes the behavioral choice throughout the stimulus, delay, and response epochs (*Figure 7C,F*). Although there are slight differences between the results of the ideal observer and targeted dimensional reduction approaches, they reveal similar overall encoding dynamics in each region. Carrying out the same analysis with the populations of suppressed neurons in each region (*Figure 5—figure supplement 1*) revealed that there was little coding of task-relevant variables, with the exception of a small amount of behavioral choice encoding in fMC (data not shown).

Taken together, these results suggest a working hypothesis of the roles that neural activity in regions V1, PPC, and fMC play in the performance of the task. Specifically, the results suggest that sensory input is primarily processed during the stimulus epoch, first in V1, then subsequently in PPC. Neural activity related to the behavioral choice arises in PPC and fMC shortly after the peak in stimulus identity coding, and is sustained in both regions throughout the delay and response epochs. This implies that stimulus identity is rapidly transformed into a behavioral choice within the stimulus epoch (possibly within PPC), and then the behavioral choice is maintained in higher regions (potentially in both PPC and fMC) until the relevant motor action is performed (*Figure 7—figure supplement 1*).

## Photoinhibition of specific cortical regions

Although trial-locked neural activity provides insight into which cortical regions may contribute to behavioral performance during different epochs of the task, it is important to note that the presence of neural activity in a region does not demonstrate a functional role in task performance. To directly test the plausibility of our working hypothesis (*Figure 7—figure supplement 1*), we next used an optogenetic inactivation approach to directly test the necessity of spatially defined cortical regions during specific epochs of the task. To inactivate the cortical regions of interest, we utilized the VGAT-ChR2-EYFP transgenic mouse line (*Zhao et al., 2011*) to rapidly and reversibly inhibit neural activity. The VGAT-ChR2-EYFP mice express the light-sensitive cation channel channelrhodopsin (ChR2) in all cortical inhibitory neuron subtypes, and thus stimulation of cortex with 473 nm light effectively silences the activity of nearby pyramidal cells (*Zhao et al., 2011*; *Guo et al., 2013*). To characterize the efficacy of optogenetic inhibition, we performed cell-attached recordings from regular-spiking neurons in L2/3 and L5 of region V1 (*Figure 8A*). We found that laser stimulation (2 s continuous pulses, 6.5 mW mm$^{-2}$) potently inhibited the activity of regular spiking neurons (see example L5 neuron, *Figure 8B*), completely eliminating spikes during the laser stimulation period in all recorded neurons, even during presentation of a drifting grating stimulus of the preferred

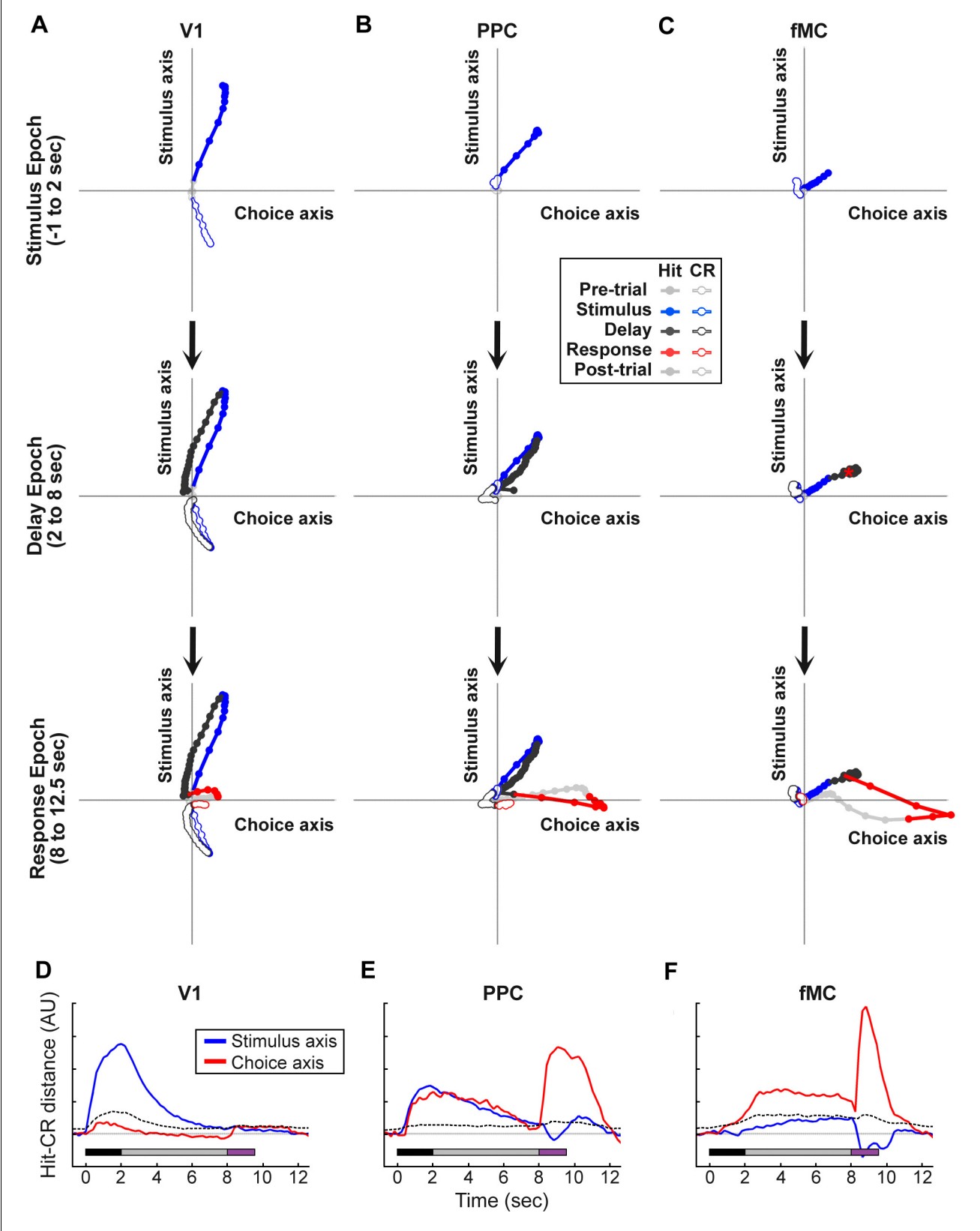

**Figure 7.** Dynamics of population responses in each region during sensorimotor decision task. Population responses were mapped onto a reduced dimensional space in which each axis corresponds to a task variable (stimulus identity, behavioral choice). The axes were defined by performing
*Figure 7 continued on next page*

*Figure 7 continued*

a regression on the principal components of the population responses for each region. (A) Hit and Correct reject (CR) response trajectories in V1 as a function of time throughout the trial. Top, during the stimulus presentation, Hit and CR trajectories diverge along the stimulus axis. Middle, during the delay period, both trajectories return to baseline. Bottom, during the response, there is a small divergence between Hit and CR trajectories along the choice axis. (B) Hit and Correct reject (CR) response trajectories in PPC as a function of time throughout the trial. Top, during the stimulus presentation, Hit and CR trajectories diverge along both the stimulus and choice axes. Most of the divergence comes from the movement of the Hit trajectory, as the CR trajectory remains close to baseline (likely due to the small number of neurons responding to non-target stimuli). Middle, during the delay period, the Hit trajectory mostly returns to baseline. Bottom, during the response, there is a second divergence between Hit and CR trajectories along the choice axis. (C) Hit and Correct reject (CR) response trajectories in fMC as a function of time throughout the trial. Top, during the stimulus presentation, Hit and CR trajectories diverge along the choice axis. As in PPC, the CR trajectory remains close to baseline. Middle, during the delay period the Hit and CR trajectories remain separated along the choice axis, suggesting sustained encoding of choice-related information (red asterisk, location of Hit trajectory at end of delay period). Bottom, during the response, there is a further divergence between Hit and CR trajectories along the choice axis. (D) The distance between the Hit and CR trajectories along the stimulus identity (blue) and behavioral choice (red) axes for region V1. This measurement is a proxy for population encoding of stimulus identity and behavioral choice (compare to *Figure 6D*, left). Dotted lines indicate upper 95% confidence interval for 1000 permutations with shuffled labels. (E,F) Same as (D) for regions PPC and fMC, respectively (compare to *Figure 6D*, middle and right).

The following figure supplement is available for figure 7:

**Figure supplement 1.** Working hypothesis of information propagation.

orientation (grating presented from 1 s before laser stimulation to 1 s after laser stimulation; *Figure 8B*, inset, $n$ = 2 mice). To characterize the spatial extent of photoinhibition, we performed immunohistochemistry using c-Fos as a marker of neural activity (*Figure 8C*). Naïve VGAT-ChR2-EYFP and control mice were implanted with windows and periodically illuminated with 2 s continuous pulses of 473 nm light (6.5 mW mm$^{-2}$). We found that after light exposure, VGAT-ChR2-EYFP mice (but not wild-type controls) showed dramatically reduced expression of c-Fos throughout all layers of the cortex underlying our cranial window (*Figure 8D,E, n* = 3 mice). The suppression was confined to within 500 µm laterally from the edge of the window, and did not spread past the white matter into subcortical regions. Similar results were achieved using an Arc antibody ($n$ = 2 mice).

Using a simplified version of the task without a delay period, we confirmed that bilateral inactivation of V1 during the stimulus epoch impaired performance of VGAT-ChR2-EYFP mice (p=0.002), but not wild-type controls exposed to the same light stimulation ($\Delta d'$ = +0.14, p=0.27; *Figure 8— figure supplement 1A–C*). In some neurons, we noticed that it took a few hundred milliseconds for the spike rate to recover to baseline (e.g., neuron in *Figure 8B*). To ensure that inactivation did not have long-lasting effects on the inactivated cortical regions, we inactivated region V1 for 2 s prior to the stimulus onset, and found no effect on performance ($\Delta d'$ = −0.15, p=0.51; *Figure 8—figure supplement 1D–F*).

After training VGAT-ChR2-EYFP mice on the task, we implanted bilateral glass cranial windows above V1 ($n$ = 4 mice), PPC ($n$ = 4), or fMC ($n$ = 4; *Figure 9B*). We tested the necessity of each region during each of the three task epochs (stimulus, delay, response) with high temporal specificity (*Figure 9A*) by randomly interleaving 'laser ON' and 'laser OFF' trials in the same behavioral session. Photoinhibition of all three regions was capable of disrupting the performance of the task, causing performance to drop to chance levels. However, the specific task epochs during which laser stimulation disrupted behavior differed for each brain region (*Figure 9C*).

As expected, V1 (*Figure 9C,D*, top row) was necessary during the stimulus epoch, with laser stimulation dramatically reducing the animal's ability to perform the task (stimulus: $d'_{OFF}$ = 1.51 ± 0.11, $d'_{ON}$: 0.21 ± 0.11, p<10$^{-3}$). However, inactivation of V1 during the delay or response epochs had no effect (delay: $\Delta d'$ = 0.47, p=0.41, response: $\Delta d'$ = 0.23, p=1.0). Activity in PPC (*Figure 9C,D*; middle row) was also necessary during the stimulus epoch (stimulus: $d'_{OFF}$ = 1.81 ± 0.16, $d'_{ON}$: 0.27 ± 0.12, p<10$^{-3}$). However, despite the high level of activity in many PPC neurons during the delay and response periods (*Figure 5C*; *Figure 3—figure supplement 2*), inactivation of PPC during these periods had no effect on behavior (delay: $\Delta d'$ = 0.03, p=1.0, response: $\Delta d'$ = 0.24, p=0.79). By contrast, suppression of fMC (*Figure 9C and D*; bottom row) during any of the three trial epochs completely abolished performance in the task (stimulus: $d'_{OFF}$ = 1.37 ± 0.10, $d'_{ON}$: −0.05 ± 0.13, p<10$^{-4}$; delay: $d'_{OFF}$ = 1.60 ± 0.10, $d'_{ON}$: 0.32 ± 0.23, p<10$^{-4}$; response: $d'_{OFF}$ = 1.60 ± 0.17, $d'_{ON}$: -0.06 ± 0.05, p<10$^{-3}$; paired *t*-test with Bonferroni correction used for all behavioral statistical tests).

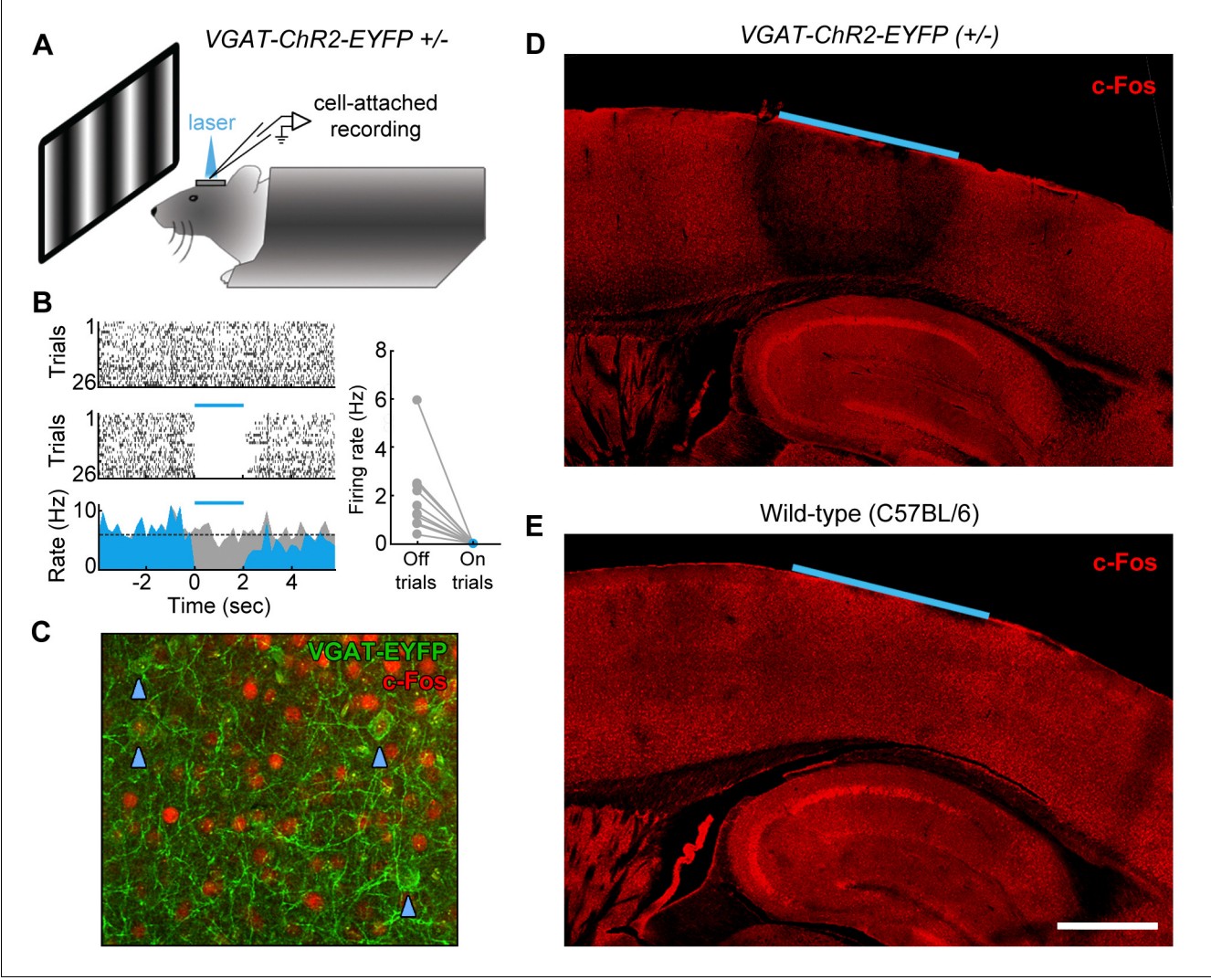

**Figure 8.** Characterization of photoinhibition in VGAT-ChR2-EYFP mice. (**A**) Schematic of cell-attached recording set-up to test the effect of blue light on regular spiking neurons in VGAT-ChR2-EYFP mice. (**B**) Response of example layer 5 regular spiking neuron with high baseline firing rate on interleaved laser off (top) and laser on (bottom) trials. Blue light completely silences neural activity during all trials (bottom). Complete silencing was observed for all regular spiking neurons recorded (right, *n* = 10 neurons). (**C**) Cortical slice from VGAT-ChR2-EYFP mouse showing constitutive c-Fos expression (red) in putative excitatory (EYFP-negative) neural somata. Note that c-Fos expresses at low levels in inhibitory interneurons (EYFP-positive; blue arrowheads). (**D**) Laser stimulation in VGAT-ChR2-EYFP mice dramatically reduces c-Fos expression (red) in a local region underneath the cranial window. Reduction of c-Fos spreads farthest in middle layers, but is generally limited to a few hundred microns from the window edge. Reduction of c-Fos was not observed in subcortical regions. The light blue line indicates window location (2 mm diameter window). (**E**) Identical laser stimulation in wild-type mice does not affect c-Fos expression (red) underneath the cranial window. The light blue line indicates window location (2 mm diameter window). Scale bar, 1 mm.

The following figure supplement is available for figure 8:

**Figure supplement 1.** Photoinhibition control experiments.

One concern is that suppression of fMC could have disrupted performance in a trivial manner by preventing the execution of motor commands. This was indeed observed for inactivation of fMC during the response period, as licking behavior was abolished during laser ON trials (*Figure 9D*, bottom right). However, prevention of motor function cannot explain decreased performance during stimulus or delay epoch inactivation, as light stimulation affected performance without decreasing lick rate (*Figure 9D*, bottom left & middle). The lick rate recovery after stimulus/delay epoch

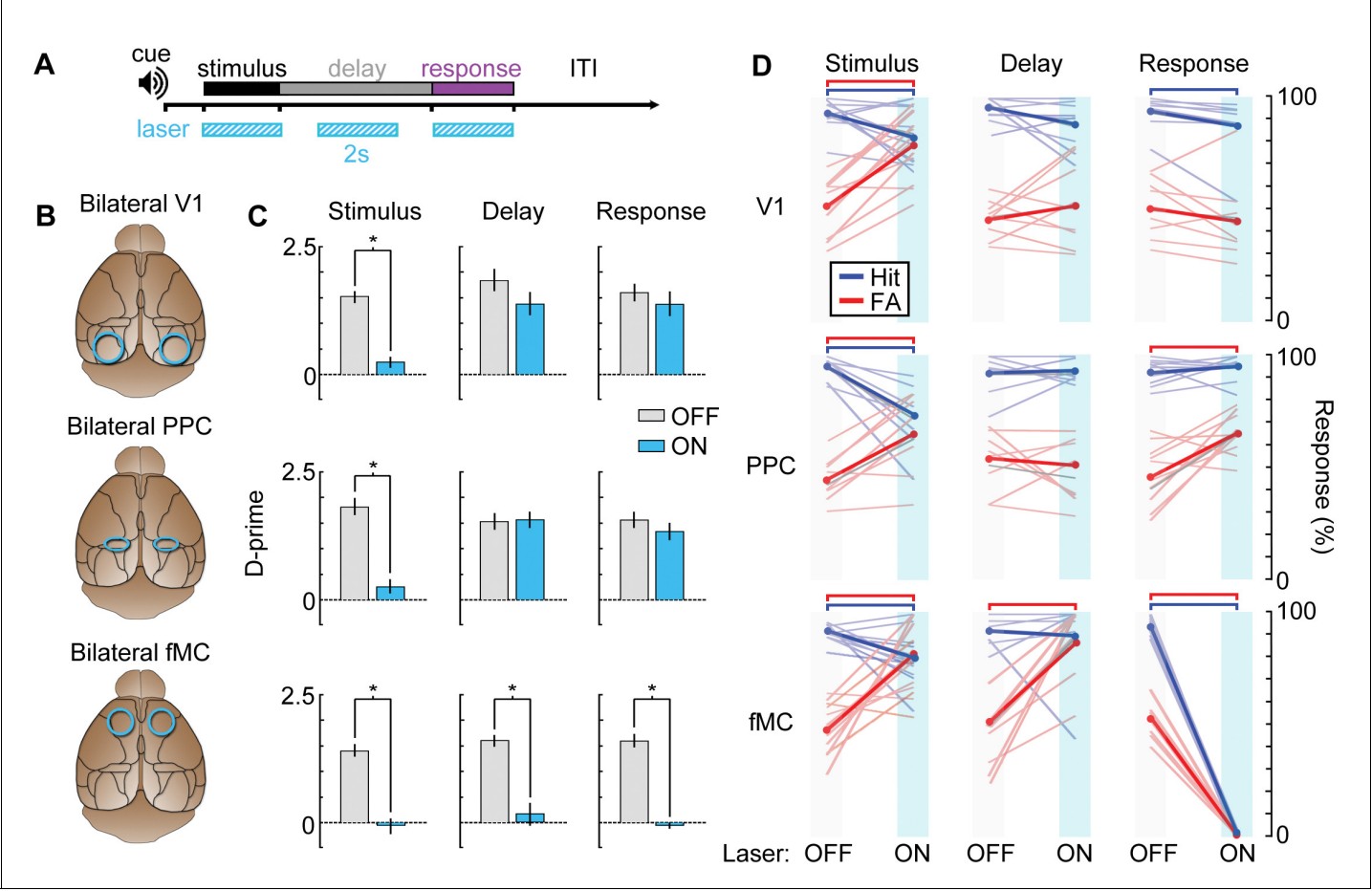

**Figure 9.** Photoinhibition of specific cortical regions during task performance. (**A**) Continuous blue light stimulation (473 nm, 2 s) was applied on interleaved trials during either the stimulus, delay, or response epochs of the task. (**B**) Glass windows were implanted bilaterally over V1 (top), PPC (middle) or fMC (bottom) of VGAT-ChR2-YFP mice. Superficial blue light stimulation silences activity in nearby pyramidal cells (*Figure 6*), effectively silencing the exposed region. (**C**) Behavioral performance (d-prime) during interleaved laser OFF and ON trials for each brain region and trial epoch. Photoinhibition of V1 (top row) or PPC (middle row) significantly disrupts behavioral performance only when applied during the stimulus period (left column), whereas photoinhibition of fMC (bottom row) during any epoch of the task significantly decreases d-prime (*\*p* < 0.05, *t*-test with Bonferroni correction). (**D**) Effect of laser stimulation on hit rate (blue) and false alarm rate (red). Lightly colored lines represent individual behavioral sessions. Colored bars at the top indicate statistical significance for each group (*p* < 0.05, *t*-test with Bonferroni correction). For all plots, error bars indicate ± S.E. M.

The following figure supplement is available for figure 9:

**Figure supplement 1.** Bilateral photoinactivation during brief (250 ms) visual stimuli.

photoinhibition is somewhat puzzling given that the vast majority of the neurons in fMC are selective for Hit trials (*Figure 5B*). In future experiments, it would be useful to image fMC activity after removal of photoinhibition to determine if the activity defaults to a 'go' activity pattern after recovery.

Another concern is that the visual stimulus duration (2 s) is likely longer than necessary to perform the discrimination, meaning that the stimulus epoch could act as a mixed stimulus/delay epoch. To test this possibility, we trained mice on a similar task with a shortened stimulus duration of 250 ms (*Figure 9—figure supplement 1A*; performance declined to chance levels with shorter stimulus durations, data not shown). Stimulus period photoinhibition continues to disrupt behavioral performance for all three regions (*Figure 9—figure supplement 1B*), similar to the longer duration stimulus epochs (*Figure 9C*). Note that recovery of activity after photoinhibition can take several hundred

milliseconds (*Figure 8B*; τ = 416 ± 91 ms), so the results should be interpreted with caution. However, these data do demonstrate that fMC is necessary either during or very shortly after the visual stimulus is shown, highlighting the rapid conversion of sensory into motor information across cortical regions.

Our results thus show that all three regions are necessary during the stimulus epoch of the task. This is in line with our imaging results (*Figure 5*) and population analyses (*Figure 6*, *Figure 7*) in showing that task-relevant information reaches frontal motor regions relatively rapidly after stimulus presentation, and suggests that PPC may be involved with converting stimulus identity representations into behavioral choice representations early in the task (during the stimulus epoch), while fMC is necessary for maintaining and executing the behavioral choice (*Figure 7—figure supplement 1*). Surprisingly, although there was robust task-related activity in PPC during the delay and response epochs, photoinhibition of PPC during these epochs had no impact on the performance of the memory-guided task. Only fMC was necessary during the delay and response epochs of the task, showing that fMC can act independently of V1 and PPC once the stimulus epoch processing is complete.

## Discussion

Understanding the circuits underlying the conversion of sensory input into motor action is a major goal of neuroscience research (*Gold and Shadlen, 2007*; *Andersen and Cui, 2009*; *Romo and de Lafuente, 2013*). By measuring and manipulating the activity of neurons in multiple cortical regions during a memory-guided sensorimotor decision task, this study has made the following contributions.

First, large-scale 2-photon calcium imaging revealed that despite differences in single-unit activity between regions, the responses within a region were surprisingly heterogeneous, with a representation of all classified response types in all regions (*Figure 3*). For example, in V1 we observed many neurons that were active during the response epoch in addition to the expected neurons active during the stimulus period, and vice versa in the frontal motor cortices. This study also shows, for the first time in mice, that a fraction of neurons (mostly in PPC and fMC) exhibit sustained activity that varies parametrically with delay duration, in line with earlier work from non-human primates (*Fuster and Alexander, 1971*; *Kojima and Goldman-Rakic, 1982*; *Funahashi et al., 1989*).

One difference between our study and many previous delayed-response studies was the use of a go/no-go design rather than an alternative choice design. Although we used this design primarily for practical reasons (to reduce training time), we found interesting biases in the cortical responses resulting from the response asymmetry (*Figure 5B*). In particular, while V1 exhibited a moderate bias toward Hit trials (65.3% target-preferring), PPC and fMC exhibited much stronger biases toward Hit trials (95.5% and 97.2% target-selective, respectively), although there was a small fraction of neurons that showed robust responses selective for CR trials (*Figure 3—figure supplements 1–3*). Given that the false alarm rate exceeded the miss rate at all delays, it is surprising that there is not more activity reflecting the suppression of licking on non-target trials. In future studies, it would be interesting to probe other regions involved in motor and cognitive control, such as striatum and prefrontal cortex, to see if there are signals corresponding to the suppression of licking during correct reject trials.

The sustained delay-period activity we observed contrasts with several recent studies in which neurons recorded in parietal and prefrontal cortices exhibited sequential neural activity (*Baeg et al., 2003*; *Fujisawa et al., 2008*; *Harvey et al., 2012*). It has been unclear whether the different modes of neural activity (steady-state vs. sequential) observed during short-term memory depend on the behavioral task, or whether the sequential activity had simply been obscured by lower-throughput methods of collecting and analyzing the data. Using volumetric calcium imaging to sample large populations of neurons, we found evidence in our task for steady-state representation of motor choice, with the vast majority of delay-sensitive neurons becoming active early in the trial and exhibiting persistent or ramping activity throughout the delay period (*Figure 2G,J,K*; *Figure 3—figure supplements 1–3*). We suspect that the nature of the mnemonic information in the behavioral task has a strong bearing on the mode of activity observed. In tasks where neurons exhibit steady-state activity, such as our study and others (*Fuster and Alexander, 1971*; *Kojima and Goldman-Rakic, 1982*; *Funahashi et al., 1989*; *Shadlen and Newsome, 2001*), the encoded motor plan is held constant throughout the delay period (e.g., lick when the spout comes forward; make a saccade to a

fixed location). In contrast, in tasks where neurons exhibit sequential activity (*Baeg et al., 2003*; *Fujisawa et al., 2008*; *Harvey et al., 2012*) the encoded motor plan may vary as a function of the trial as the animal navigates through real or virtual space. Both types of memory are important for understanding complex behavior, and warrant further study of the underlying neural architecture.

A second contribution is the finding that a sizable fraction of the significantly responding neurons (32.0% of all significantly-responding neurons) exhibited suppressed rather than enhanced responses during task performance (*Figure 2*, *Figure 3*). However, these neurons were generally much less selective than enhanced neurons (particularly in V1 and PPC), and as a result appear to play little role in encoding task-relevant variables (*Figure 4*). Without a method for specifically manipulating the activity of the suppressed neurons, it is difficult to unambiguously discern what role they may play in task performance. One possibility is that suppression of spontaneously active neurons may reduce any ongoing task-irrelevant activity, and thereby improve the readout of task-relevant population activity by higher-order neurons.

Third, we found that despite the heterogeneity observed in the single-unit activity, when the populations of neurons within each region are analyzed as a whole, they exhibit distinct coding dynamics of task-relevant variables (*Figure 6*, *Figure 7*). Specifically, the V1 population predominantly encodes the stimulus identity, particularly early in the trial. Similar to previous research in non-human primates (*Park et al., 2014*) and rodents (*Raposo et al., 2014*), we find multiplexed encoding of stimulus identity and behavioral choice in PPC. However, we find that the dynamics of the stimulus identity and behavioral choice encoding in PPC are distinct, with choice encoding lagging slightly, and lasting throughout the delay period. The fMC predominantly encodes behavioral choice, beginning early in the trial and lasting throughout the delay period until after the motor response is made, in accord with choice-related activity observed in premotor regions in earlier rodent (*Erlich et al., 2011*; *Guo et al., 2013*, *Li et al., 2015*) and non-human primate studies (*Hernandez et al., 2010*).

Fourth, we find that PPC is necessary for performance during the stimulus epoch of the task (*Figure 9*), in contrast to several recent pharmacological and optogenetic perturbation studies in rodents (*Guo et al., 2013*; *Erlich et al., 2015*), which found that inactivation of PPC had no effect on behavioral performance in sensorimotor decision tasks. Why does PPC appear to play a critical role in some tasks but not in others? One possibility is that PPC plays a greater role for some sensory modalities than others during sensorimotor decision tasks, perhaps due to differences in anatomy (e. g., S1 directly projects to motor regions while V1 input reaches motor cortex only indirectly, with PPC as the principal cortical intermediary; *Wang et al., 2012*). Indeed, a recent study found that pharmacological inhibition of PPC somewhat impaired performance of a rate discrimination task for visual stimuli, but did not affect performance when auditory stimuli were used (*Raposo et al., 2014*). Another intriguing possibility is that PPC plays a greater role for sensorimotor tasks in which higher-level features (e.g., orientation) must be mapped to motor actions; in contrast to tasks involving detection of simple stimuli, which may be mediated by subcortical or direct sensory cortex to motor cortex projections. This could be tested directly by investigating the necessity of PPC in tasks that differ in the complexity of the discriminated feature while the rest of the task structure is kept constant.

Finally, this study is in accord with several recent rodent studies in finding that fMC, but not PPC, appears to be responsible for maintenance of the motor plan during the delay period (*Hernandez et al., 2010*; *Erlich et al., 2011*; *Guo et al., 2013*; *Li et al., 2015*). Indeed, while photoinhibition of fMC during the delay period results in complete disruption of behavioral performance, photoinhibition of PPC during the delay period has no significant effect (*Figure 9*). This suggests that, at least in motor planning tasks, PPC is likely more important for mapping stimulus identity to behavioral choice than for maintaining the choice in short-term memory stores. A separate study from our lab using a zero-delay version of the task revealed a large number of spatially-intermixed neurons in PPC (but not V1) encode the response choice in addition to the stimulus, and will actually switch selectivity if the target and non-target stimuli are switched (*Pho et al., 2015*). However, additional research will be necessary to determine if the PPC is indeed the locus for stimulus-response mapping in these tasks. Another intriguing finding is that the dependency on fMC arises quite early in the task (*Figure 9*), within hundreds of milliseconds from stimulus onset (*Figure 9—figure supplement 1*). This result appears to conflict with earlier findings using unilateral inactivation of motor cortex (*Guo et al., 2013*; *Hanks et al., 2015*), but is consistent with a recent study finding that delay-period motor cortical activity can be recovered if the contralateral hemisphere is spared (*Li et al.,*

*2016*). The early dependency on fMC activity suggests a very rapid conversion of stimulus information into a motor plan.

An important next step is to determine the role of V1, PPC, and fMC in a task that requires memory of the stimulus, such as a delayed match-to-sample task (*Miller et al., 1996*; *Romo et al., 1999*; *Liu et al., 2014*). It seems probable that in such tasks, PPC or perhaps even primary sensory cortex would play a greater role during delay-period maintenance. Given the importance of both sensory and motor short-term memory, both delayed-response and delayed-comparison tasks warrant further study.

A few caveats are important to note in the interpretation of this study. First, some care must be taken in the interpretation of inactivation experiments, both in our study and previous studies (*Guo et al., 2013*; *Erlich et al., 2015*; *Raposo et al., 2014*). It should be noted that if optogenetic or pharmacological inhibition in a given area impairs performance, it does not prove that local processing in the inactivated region is playing a crucial role. Instead, it could be that the impairment is mediated by feedback projections from the inhibited region to other parts of the cortex or subcortical structures. For example, the behavior deficit seen during stimulus epoch photoinhibition of PPC or fMC could be due to a lack feedback from fMC to visual, parietal, or other (sub)cortical regions, perhaps disrupting network homeostasis (*Otchy et al., 2015*). This is particularly a worry when trying to assess the independent role of highly interconnected and/or adjacent cortical regions (e.g., visual and parietal cortices). In future studies, simultaneous imaging and inactivation will help ensure that optical perturbations are restricted to the desired region.

Second, one persistent issue in delayed-response tasks (in which the correct response is evident early in the trial) is that the subject could adapt a behavioral strategy in which they make a micropostural preparation (e.g., movement of the eyes, changes in body orientation or tongue position) to stand in for the eventual motor action. This in turn may preclude the necessity for actively maintaining a motor plan throughout the delay period. This problem has been recognized for some time (*Kojima and Goldman-Rakic, 1982*), but it remains challenging to control for the wide range of potential covert movements. Disentangling the effects of covert preparations on premotor and motor activity will be an important step in understanding the neural basis of motor planning.

Finally, this study focused on cortical regions located on the dorsal surface of the mouse brain, as they are readily accessible to both 2-photon calcium imaging and optogenetic inhibition. However, there are likely to be additional cortical and subcortical regions involved in this task, including prefrontal cortex (*Fuster and Alexander, 1971*; *Kojima and Goldman-Rakic, 1982*; *Funahashi et al., 1989*), thalamus (*Fuster and Alexander, 1973*), superior colliculus (*Kopec et al., 2015*), and basal ganglia (*Kawagoe et al., 1998*; *Ding and Gold, 2012*). These regions could be further investigated with a similar approach as invasive (*Barretto and Schnitzer, 2012*; *Zorzos et al., 2012*; *Andermann et al., 2013*) and noninvasive (*Filonov et al., 2011*; *Mittmann et al., 2011*; *Prakash et al., 2012*) techniques for optical interrogation of deeper structures become available.

## Materials and methods

### Surgical procedures

All procedures were approved by the Massachusetts Institute of Technology Animal Care and Use Committee. Data were collected from male adult (60–120 day old) wild-type (RRID:IMSR_JAX: 000664; *n* = 12) and VGAT-ChR2-EYFP mice (RRID:IMSR_JAX:014548; *n* = 15). The animals were housed on a 12 hr light/dark cycle in cages of up to 5 animals before the implants, and individually after the implants. All surgeries were conducted under isoflurane anesthesia (3.5% induction, 1.5–2.5% maintenance). Meloxicam (1 mg kg$^{-1}$, subcutaneous) was administered pre-operatively and every 24 hr for 3 days to reduce inflammation. Once anesthetized, the scalp overlying the dorsal skull was sanitized and removed. The periosteum was removed with a scalpel and the skull was abraded with a drill burr to improve adhesion of dental acrylic. Stereotaxic coordinates for future viral injections were marked with a non-toxic ink and covered with a layer of silicon elastomer (Kwik-Sil, World Precision Instruments) to prevent acrylic bonding. The entire skull surface was then covered with dental acrylic (C&B-Metabond, Parkell) mixed with black ink to reduce light transmission. A custom-designed stainless steel head plate (eMachineShop.com) was then affixed using dental

acrylic. After head plate implantation, mice recovered for at least five days before beginning water restriction.

After behavioral training was complete, animals were taken off water restriction for five days before undergoing a second surgery to implant the imaging/photoinhibition window(s). Procedures for anesthetic administration and post-operative care were identical to the first surgery. The dental acrylic and silicon elastomer covering the targeted region were removed using a drill burr. The skull surface was then cleaned and a craniotomy (2–4 mm, depending on targeted structure) was made over the region of interest, leaving the dura intact. For imaging experiments, neurons were labeled with a genetically-encoded calcium indicator by microinjection (Stoelting) of 50 nl AAV2/1.Syn. GCaMP6s.WPRE.SV40 (University of Pennsylvania Vector Core; diluted to a titer of $10^{12}$ genomes ml$^{-1}$) 300 μm below the pial surface. Between two and five injections were made in each exposed region, centered at V1 (4.2 mm posterior, 2.5 mm lateral to Bregma), PPC (2 mm posterior, 1.7 mm lateral to Bregma) or fMC (1.5 mm anterior, 1 mm lateral to Bregma). Since the viral expression spreads laterally from the injection site, exact stereotaxic locations were photographed through the surgical microscope for determining imaging areas. For photoinhibition experiments, craniotomies were made bilaterally, with no virus injection. Finally, a cranial window was implanted over the craniotomy and sealed first with silicon elastomer then with dental acrylic. The cranial windows were made of two rounded pieces of coverglass (Warner Instruments) bonded with optical glue (NOA 61, Norland). The bottom piece was circular or oval, custom cut according to cortical region(s) (V1: $2.5 \times 2.5$ mm; PPC: $1 \times 2$ mm; V1 + PPC: $4 \times 4$ mm; fMC: $2 \times 2.5$ mm, anterior-posterior x medial-lateral) and fit snugly in the craniotomy. The top piece was a larger circular coverglass (3–5 mm, depending on size of bottom piece) and was bonded to the skull using dental acrylic. Mice recovered for five days before commencing water restriction.

## Behavior

Mice were head-fixed using optical hardware (Thorlabs) and placed in a polypropylene tube to limit movement. Spout position was controlled by mounting the spout apparatus on a pressure-driven sliding linear actuator (Festo) controlled by two solenoids (Parker). Licks were detected using an infrared emitter/receiver pair (Digikey) mounted on either side of the retractable lick spout. Rewards consisted of 5–8 μl water and punishments consisted of a white noise auditory stimulus alone (early training) or white noise plus 1–3 μl of 5 mM quinine hydrochloride (Sigma) in water (late training). Behavioral training and testing was implemented with custom software written in Matlab (Mathworks). Drifting grating stimuli were presented with the Psychophysics Toolbox (*Brainard, 1997*). Mice were trained during the light cycle. The stimulus consisted of sine wave gratings (spatial frequency: 0.05 cycles deg$^{-1}$; temporal frequency: 2 Hz) drifting at either 0 degrees (target) or 90 degrees (non-target) away from vertical. These stimuli were chosen to drive distinct groups of visual neurons with roughly equal strength (*Figure 1—figure supplement 2*). Thus, any large differences in stimulus selectivity observed in cortical neurons are not likely the result of stimulus strength.

Mice were trained in successive stages, with advancement to the next stage contingent on correct performance: 1) Mice received reward any time they licked the spout. 2) Trial structure was initiated by having an auditory cue tone, followed by a visual stimulus (100% targets), followed by an inter-trial interval. Mice were only rewarded for licks during the visual stimulus. 3) Once mice exhibited preferential licking during the stimulus compared to inter-trial interval, the target rate was reduced over several sessions from 100% to 50%. At this point, the non-target was a static grating orientated orthogonally to the target. Licks during non-targets were punished with white noise or white noise plus quinine. 4) Once mice exhibited the ability to discriminate target and non-target gratings ($d'$ > 1 and $R_{HIT}$ - $R_{FA}$ > 30% for consecutive sessions, where $R_{HIT}$ and $R_{FA}$ are the hit and false alarm rate, respectively), the temporal frequency of the non-target grating was increased. 5) Spout withdrawal was introduced. At first the spout was extended within range before the stimulus appeared, then spout extend time was gradually delayed until after the stimulus had turned off (i.e., 0 s delay). 6) Finally, the variable delay period was gradually increased to 0/3/6 s (imaging mice) or 0/2/4 s (photoinhibition mice). Mice that failed to fully learn the task within 150 sessions or showed signs of infection were removed from the study. In total, we removed 11 mice from the study before data collection was complete: 3 mice for failure to consistently lick the spout for reward (stage 1), 3 mice for failure to progress during the visual discrimination phase (stage 3), 1 mouse for failure to

progress at the variable delay stage (stage 6), 1 mouse that showed signs of infection, and 4 mice that completed behavioral training but either had poor viral expression or cloudy windows after surgery.

Once mice reached high levels of performance at the final stage of the task ($d' > 1.5$ and $R_{HIT}$ - $R_{FA} > 50\%$), they were removed from water restriction for window implantation. After recovery from window implantation surgery, they were re-trained to a level of high performance (2–7 days) before beginning experimental sessions. For both imaging and photoinhibition experiments, any sessions with poor performance were discarded (minimum performance criterion: $d' > 1$ and $R_{HIT}$ - $R_{FA} >$ 30%). For photoinhibition experiments, the performance criterion was applied to the control condition.

For analysis of movement during the delay period (*Figure 1—figure supplement 1*), cropped video frames (300 × 200 pixels; width x height) from Hit and CR trials were compared to a 'template' CR image to measure postural changes or changes (increases or decreases) in movement during each epoch of the task (Pre-stimulus, Delay, Response). Since some amount of movement is expected in all conditions, a pixel-wise map of the absolute difference between single CR frames and the CR template within-condition ($D_{CR}$) was calculated as a measure of baseline movement:

$$D_{CR}(x,y) = |CR_f(x,y) - \overline{CR_{F \neq f}(x,y)}|$$

Where $f$ is the index of a single CR frame and $F \neq f$ is the set of all CR frames except $f$, and where $x$ and $y$ are pixel indices. The absolute difference map was calculated separately for each epoch (Pre-stimulus, Delay, Response). A pixel-wise map of the absolute difference between single Hit frames and the CR template ($D_{Hit}$) was calculated in the same manner:

$$D_{Hit}(x,y) = |Hit_f(x,y) - \overline{CR_{F \neq f}(x,y)}|$$

In cases where the number of Hit frames exceeded the number of CR frames, the excluded frame was chosen at random from the CR frames. Finally, the difference in movement on Hit trials relative to CR trials ($D_{Sub}$) was calculated by taking the absolute value of the subtracted difference maps (*Figure 1—figure supplement 1B*):

$$D_{Sub}(x,y) = |D_{Hit}(x,y) - D_{CR}(x,y)|$$

Note that since the frames are compared against a CR template, this approach will capture not only transient movement but also stable postural changes specific to Hit trials. To compare between sessions, the subtracted difference maps ($D_{Sub}$) were averaged across all pixels for each epoch (*Figure 1—figure supplement 1C*).

## 2-Photon imaging

GCaMP6s fluorescence was imaged 14–35 days after virus injection using Prairie Ultima IV 2-photon microscopy system with a resonant galvo scanning module (Bruker). For fluorescence excitation, we used a Ti-Sapphire laser (Mai-Tai eHP, Newport) with dispersion compensation (Deep See, Newport) tuned to $\lambda = 910$ nm. For collection, we used GaAsP photomultiplier tubes (Hamamatsu). To achieve a wide field of view, we used a 16x/0.8 NA microscope objective (Nikon), which was mounted on a Z-piezo (Bruker) for volume scanning. Resonant scanning was synchronized to z-piezo steps in the acquisition software for volume scanning. For volume scanning, four 441 × 512 pixel imaging planes separated by 20 or 25 µm were imaged sequentially at a stack rate of 5 Hz for 10 min imaging sessions. Occasionally, very bright neurons were visible across multiple planes. To exclude redundant sampling of the same neuron, the Pearson correlation coefficients of the fluorescence traces of all pairs of neurons within a recording were calculated. Neuron pairs with a correlation coefficient > 0.5, and an inter-ROI distance <12.5 µm in the XY plane were considered to be redundant and the ROI with the lower average fluorescence signal (more likely out of plane) was removed. Even with virus titer dilution, a small number of nucleus-filled neurons were observed in most experiments (*Chen et al., 2013*), but they comprised a small percentage of neurons and generally did not exhibit significant task-driven responses (*Harvey et al., 2012*). Laser power ranged from 40–75 mW at the sample depending on GCaMP6s expression levels. Photobleaching was minimal (<1% min$^{-1}$) for all laser powers used. A custom stainless steel plate (eMachineShop.com) attached to a black curtain

was mounted to the head plate before imaging to prevent light from the visual stimulus monitor from reaching the PMTs. During imaging experiments, the polypropylene tube supporting the mouse was suspended from the behavior platform with high tension springs (Small Parts) to dampen the movement.

## Image analysis

Images were acquired using PrairieView acquisition software and sorted into multi-page TIF files. All subsequent analyses were performed in MATLAB (Mathworks). First, images were corrected for X-Y movement by registration to a reference image (the pixel-wise mean of all frames) using 2-dimensional cross correlation. Movements in the Z-dimension were rare in normal imaging conditions, although movements were sometimes observed during licking. In order to prevent licking-related artifacts from being identified as significant responses, we adapted our response significance test to exclude short-duration changes in fluorescence (see Analysis of task-driven responses section).

To identify responsive neural somata, a pixel-wise activity map was calculated as previously described (*Ahrens et al., 2012*). Neuron cell bodies were identified using local adaptive threshold and iterative segmentation. Automatically-defined ROIs were then manually checked for proper segmentation in a MATLAB-based graphical user interface (allowing comparison to raw fluorescence and activity map images). To ensure that the response of individual neurons was not due to local neuropil on somatic signals, a corrected fluorescence measure was estimated according to previously described methods (*Chen et al., 2013*) as $F_{corrected\_soma}(t) = F_{raw\_soma}(t) - 0.7 \times F_{neuropil}(t)$, where $F_{neuropil}$ was the defined as the fluorescence in the region 0–15 μm from the ROI border (excluding other ROIs). The $\Delta F/F$ (corrected and uncorrected) for each neuron was calculated as $\Delta F/F_t = (F_t - F_0)/F_0$, with $F_0$ defined as the mode of the raw fluorescence density distribution.

## Analysis of task-driven responses

After image preprocessing and $\Delta F/F$ extraction, traces were sorted into matrices by trial type (hit, miss, correct reject, false alarm). Testing neurons for significant responses was complicated by the large number of neurons (>9000) and the number of samples per trial. Using a traditional threshold approach (with responses considered significant if they pass a threshold of multiple SDs from baseline) would yield either an unreasonable number of false positives (if a low threshold was used) or an unreasonable number of false rejections (if a high threshold was used). The GCaMP6s indicator has a long decay time constant ($\tau_{1/2} > 1$ s), with fluorescence transients staying above baseline for >3 s for even a single action potential (*Chen et al., 2013*). Thus, to capture weak but reliable trial-locked activity while excluding artifacts, we used a low significance threshold (sample different from baseline at p<0.05, Wilcoxon signed-rank test) but required at least 10 samples to be significant in the same direction for at least 2 (of 3) delays for either hit or CR trials to be considered statistically significant (probability of significant response for any one neuron purely by chance $<10^{-9}$). Since reliable GCaMP6s signals will exhibit slow decay over 10 or more samples, this approach will allow genuine calcium signals to emerge as significant, while chance fluctuations in fluorescence and short-duration movement artifacts (such as during licking) will not pass the significance threshold. All neurons that were significantly different from baseline according to these criteria in either the positive (enhanced activity) or negative (suppressed activity) were included in further analysis. To ensure the neuropil correction procedure did not create artificial responses, we required that both uncorrected $\Delta F/F$ and corrected $\Delta F/F$ exhibit significant changes for neurons to be included. Once neurons were determined to have significant responses, the corrected $\Delta F/F$ was used for all further analyses. For some analyses, the $\Delta F/F$ responses were normalized by subtracting the baseline response (1 s before stimulus onset) and dividing by the maximum (for enhanced neurons) or minimum (for suppressed neurons) trial-averaged $\Delta F/F$ for the neuron.

To investigate the clustering of task-driven responses (*Figure 3*), we first de-noised and reduced the dimensionality of the data by taking the first 20 principal components (explaining 98% of the variance). To cluster the data, we computed linkages using Ward's method (with Euclidean distance). The dendrogram revealed distinct groups with high inter-cluster distance (*Figure 3A*). Averaging the responses of neurons within these clusters yielded distinct response types that corresponded well to the range of observed responses (*Figure 3B*). To determine the number of clusters ($K = 6$), we iteratively increased the number of clusters until the average traces began showing highly overlapping

responses (at $K > 6$). Manual inspection revealed a small number of neurons within each cluster that appeared to be miscategorized. To improve categorization (*Figure 3C*), we computed the Pearson correlation coefficient between the response of each neuron and the mean of each of the six response types. Neurons were re-categorized based on the response type with the highest correlation to their average response, yielding distinct clusters with high face validity and internal consistency (*Figure 3—figure supplement 1–3*).

The trial selectivity index (*Figure 4C*) was computed as $SI = (R_{pref} - R_{non-pref})/(R_{pref} + R_{non-pref})$, where $R_{pref}$ and $R_{non-pref}$ are the responses to the preferred and non-preferred trials, respectively. The modal latency of the population (*Figure 4D*, *Figure 5D*) was estimated by first taking the DF/F of each neuron, and upsampling (via linear interpolation) the mean $\Delta F/F$ signal to 1000 Hz. We then determined the point at which the mean $\Delta F/F$ signal for each neuron went above or below baseline in a sustained manner ($>1\sigma$ from mean baseline $\Delta F/F$ for 1000 consecutive samples, or 1 s in trial time) and used the first sample as the estimated latency for that neuron. Since the latency distribution of the population was highly skewed in higher regions, we used the first mode of the latency density distribution to describe the onset of population activity (full latency distributions for 6 s delay trials shown in *Figure 5D*). Note that these latencies are computed on upsampled data based on a linear interpolation, so the values should be considered approximate. To measure whether the neurons within a region exhibited delay modulation (*Figure 4E*) we fit the slope of the integrated $\Delta F/F$ across increasing delay durations using a first order polynomial. We then calculated the delay modulation index (DMI) as $DMI = (n_{pos}-n_{neg})/(n_{pos}+n_{neg})$ for enhanced populations and $DMI = (n_{neg}-n_{pos})/(n_{pos}+n_{neg})$ for suppressed populations, where $n_{pos}$ is the number of neurons with a positive slope and $n_{neg}$ is the number of neurons with a negative slope.

## Population encoding of task-related variables

In order to determine how well we could decode the stimulus identity for populations of neurons (*Figure 6B*), we measured the discriminability of population responses on error trials from correct trials with a different stimulus but the same response. For example, to compare miss and correct reject (CR) population responses (different stimuli, same response), for each time point we first calculated the Euclidean distances between the neural population response vector at each CR trial *t* and a template CR response vector (average of all CR trials except *t*) and calculated the Euclidean distances between each miss trial *t* and the template CR response (average of all trials). We then generated a receiver operating characteristic curve from the distances to determine the discriminability of the miss from CR responses. By taking the area under the receiver operating characteristic (auROC; *Britten et al., 1992*) for each time point, we could quantify the performance of an ideal observer in discriminating stimulus identity based purely on the population responses as a function of time during the trial. For behavioral choice encoding (*Figure 6C*), the same analysis was carried out comparing error-correct trial pairs with the same stimulus but different response. To reduce noise from experiments with very few error trials, only experiments that had at least three error trials of each type (miss, false alarm) were included in the analysis. To determine how coding varied as a function of neural population size, we averaged the performance across 1000 iterations using populations of 1, 5, 10, 50, or 100 randomly selected neurons (*Figure 6D*). For significance testing, we calculated the auROC for 1000 permutations with shuffled trial labels. Individual time points were considered to predict stimulus/choice if the mean auROC was greater than the 95% confidence interval of the shuffled permutations for at least 3 consecutive time points (to compensate for multiple comparisons).

Targeted dimensionality reduction (*Figure 7*) was performed as previously described (*Mante et al., 2013*). Briefly, we first used principal components analysis to obtain a 'de-noised' estimate of the most prominent response patterns in the population. We then used linear regression to define de-mixed task-related axes of stimulus (y-axis) and choice (x-axis) from the principal components. Average population responses in the two-dimensional state space were plotted for hit and CR trials as a function of time throughout the trial. To reduce noise from experiments with very few error trials, only experiments that had at least three error trials of each type (miss, false alarm) were included in the analysis. To quantify population coding along the stimulus and choice dimensions (*Figure 7D*), we simply subtracted the CR trajectory from the hit trajectory along the stimulus (blue) and choice (red) axes. Chance difference levels were determined by calculating the 95% confidence interval for 1000 permutations with shuffled trial labels.

## Electrophysiology methods

For in vivo anesthetized electrophysiology experiments (*Figure 8A,B*), we used procedures similar to those previously described (*Wilson et al., 2012*). Briefly, transgenic VGAT-ChR2-EYFP mice (n = 2) were anesthetized with isoflurane (1.5%), with body temperature maintained at 37.5°C with a heating blanket. A metal head plate was attached to the skull using superglue and dental acrylic, and a 1 mm craniotomy was performed over V1. The dosage of anesthesia was then lowered (1%) before beginning electrophysiology.

Recordings were made using custom software written in Matlab (Mathworks) controlling a Multi-Clamp 700B Amplifier (Axon) that measured differences between a glass pipette electrode inserted into the brain at 27° and an Ag/AgCl ground pellet electrode positioned in the same solution as the brain. Borosilicate pipettes (outer diameter = 1.5 mm, inner diameter = 1.17 mm) were pulled using a Sutter P-2000 laser puller (Sutter Instruments) to a diameter corresponding to 3–7 MΩ. The pipette was back-filled with Alexa Fluor 488 (Molecular Probes) and targeted to the injection site using a 10x lens. Cells were targeted blindly by advancing diagonally through the cortex with light positive pressure. Cell proximity was detected through deflections in electrical resistance observed in voltage clamp during a repetitive 5 mV command pulse. Once resistance had increased a few MΩ, slight negative pressure was applied and the pipette was advanced more slowly until resistance further increased (to a final value of 10–30 MΩ) and/or spikes were detected visually or via an audio monitor. At that point, the amplifier was switched to current clamp to record spikes.

Optogenetic activation of local inhibitory cells (*Figure 8*, *Figure 9*) was achieved using a 200 mW 473 nm diode-pumped solid state blue laser (Opto-Engine) coupled with a 200 μm fiber. Laser intensity was modulated with a variable neutral density filter (Thorlabs) to match the intensity used during behavioral experiments (6.5 mW mm$^{-2}$). Full-field drifting grating stimuli were presented for 4 s, with each presentation preceded by a 6 s 'off' period with gray screen. Optogenetic stimulation occurred on every other stimulus presentation during the middle 2 s of the 'on' period. Data was analyzed using Matlab, with spikes detected using a manually-defined thresholds.

## Photoinhibition

Blue light illumination was provided by a 200 mW 473 nm diode-pumped solid state laser (MBL-III-473, Opto-Engine). The laser was passed through a 50/50 splitter (CM1-BS013, Thorlabs), with each output passed through an adjustable neutral density filter (Thorlabs) into a fiber launch (PAF-X-11-PC-A, Thorlabs). Fibers terminated in ceramic ferrules that were precisely positioned above the cranial window with optical hardware. A light shield (eMachineShop.com) was attached to the head plate to prevent reflected laser light from reaching the retina and influencing behavior. Laser power was controlled by analog inputs sent from the behavior computer. Continuous light pulses (2.2 s, 6.5 mW mm$^{-2}$) were used to suppress activity in a sustained manner at low laser power (*Zhao et al., 2011*). We measured the spread of inhibition using immunohistological labeling of activity-driven protein expression (see Histology section). For photoinhibition experiments, laser stimulation was given on 50% of trials with 4 s delay (25% of all trials; no photoinhibition on 0 s or 2 s delay trials) in a randomly inter leaved manner. Laser stimulation was applied throughout the duration of visual stimulus presentation (stimulus epoch), in the middle 2 s of delay (delay epoch), or during the spout extension (response epoch).

## Histology

For photoinhibition characterization experiments (*Figure 8*), naïve VGAT-ChR2-EYFP or wild-type mice were implanted with a head plate and a 2 mm diameter cranial window over the posterior cortex (V1 or PPC). After recovery and habituation, mice were head-fixed and stimulated with 2 s light pulses (6.5 mW mm$^{-2}$) every 10 s. Drifting grating stimuli were played during the light pulses to drive neural activity. After 1.5 hr of light stimulation, mice were removed from the rig and perfused with 4% paraformaldehyde. The brain was removed and placed in 4% PFA for 24 h, followed by 24 hr in PBS. 50 μm sagittal slices were cut and placed in PBST (0.2% Triton X-100 in PBS) with 5% normal goat serum for 1 hr. They were then incubated with primary antibody at 4°C for 24 hr (rabbit Anti-c-Fos 1:200, SC-52, Santa Cruz Biotechnology, Inc.; rabbit Anti-Arc 1:200, ab118929, Abcam). After washing in PBST, slices were incubated 1 hr with secondary antibody (1:200 AlexFlour555 anti-rabbit, Invitrogen). Slices were washed again in PBST, mounted and coverslipped. Slices were imaged

on a laser scanning confocal microscope (Zeiss). Repeating this procedure with Arc as an activity marker (1:500 anti-Arc, Abcam) yielded qualitatively similar results as c-Fos (data not shown).

## General statistics

Data groups were tested for normality using the Kolmogorov-Smirnov test and then compared with the appropriate tests (t-tests, Wilcoxon rank-sum or Wilcoxon signed-rank tests, all two sided). Bonferroni correction was used for multiple comparisons. Bootstrap estimates of s.e.m. were calculated as the standard deviation of values evaluated in 1,000 bootstrap iterations, obtained by randomly re-sampling with replacement from the original values. Due to very large sample sizes, very small p-values ($<10^{-9}$) were approximated as $p<10^{-9}$ as a lower bound on reasonable probabilities. Sample sizes were not explicitly estimated, as even a single session generally had sufficient samples for the statistical tests used. However, to ensure that results were replicable between sessions and mice, we included 4–6 mice for each region in both imaging and inactivation experiments.

## Acknowledgements

We thank J Sharma, S Ramirez, B Crawford, A Boesch, V Li, C Le, and T Emery for technical assistance; S El-Boustani and R Huda for comments on the manuscript. LL Looger, J Akerboom, DS Kim, and the Genetically-Encoded Neuronal Indicator and Effector (GENIE) Project at Janelia Farm Research Campus Howard Hughes Medical Institute for generating and characterizing GCaMP6 variants; H Robertson, Q Chen and G Feng for providing VGAT-ChR2-EYFP mice.

## Additional information

### Funding

| Funder | Grant reference number | Author |
|---|---|---|
| National Institute of Neurological Disorders and Stroke | U01-NS090473 | Mriganka Sur |
| National Eye Institute | R01-EY007023 | Mriganka Sur |
| National Science Foundation | BRAIN-EAGER | Mriganka Sur |
| National Institute of Mental Health | K99-MH104259 | Michael J Goard |
| National Eye Institute | F32-EY023523 | Michael J Goard |
| National Science Foundation | Graduate Research Fellowship | Gerald N Pho |

The funders had no role in study design, data collection and interpretation, or the decision to submit the work for publication.

### Author contributions

MJG, Conception and design, Acquisition of data, Analysis and interpretation of data, Drafting or revising the article; GNP, Acquisition of data, Analysis and interpretation of data, Drafting or revising the article; JW, Acquisition of data, Drafting or revising the article; MS, Conception and design, Drafting or revising the article

### Author ORCIDs

Michael J Goard, http://orcid.org/0000-0002-5366-8501

### Ethics

Animal experimentation: This study was performed in strict accordance with the recommendations in the Guide for the Care and Use of Laboratory Animals of the National Institutes of Health. All experimental procedures were approved by the Massachusetts Institute of Technology Institutional Animal Care and Use Committee (protocol number 1014-111-17).

## Additional files

### Major datasets

The following dataset was generated:

| Author(s) | Year | Dataset title | Dataset URL | Database, license, and accessibility information |
|---|---|---|---|---|
| Michael J Goard, Gerald N Pho, Jonathan Woodson, Mriganka Sur | 2016 | Data from: Distinct Roles of Visual, Parietal, and Frontal Motor Cortices in Memory-guided Sensorimotor Decisions | http://dx.doi.org/10.5061/dryad.km140 | Available at Dryad Digital Repository under a CC0 Public Domain Dedication |

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
