## [Decision Letter]

[Editors’ note: this article was originally rejected after discussions between the reviewers, but the authors were invited to resubmit after an appeal against the decision.]

Thank you for submitting your work entitled "Distinct Roles of Visual, Parietal, and Frontal Motor Cortices in Memory-guided Sensorimotor Decisions for consideration by *eLife*. Your article has been reviewed by four peer reviewers, one of whom is a member of our Board of Reviewing Editors, and the evaluation has been overseen by Timothy Behrens as the Senior Editor. Our decision has been reached after consultation between the reviewers. Based on these discussions and the individual reviews below, we regret to inform you that your work will not be considered further for publication in *eLife*.

The following individuals involved in the review of your submission have agreed to reveal their identity: Jeffrey Erlich and Adam Kampff.

First, we are very sorry for the delay on this manuscript. It took us a long time, because we had difficulties in figuring out what to do, and we needed to recruit a fourth reviewer. We hope you will understand why when you read the reviews below. There is one reviewer who is strongly on your side, and who defended the manuscript vigorously during the first round of discussions, but the two other reviewers thought that the task design made the data interpretation difficult and therefore thought that the neurobiological conclusions that could be drawn were of interest, but should be published in a more specialised journal. When we asked a fourth reviewer, they agreed with the two more negative reviewers. The major criticism, which was the focus of most of the extensive discussion between reviewers, is perhaps most eloquently explained in the comments of reviewer 4.

*Reviewer #1:*

In order to understand the role of three distinct cortical regions in a sensorimotor transformation, Goard et al. recorded from V1, PPC and fMC of mice performing a visual discrimination task with a response delay introduced by removing the lick spout for varying times.

A number of findings from this study are re-establishing known findings. This includes findings about (i) V1: visually responsive and required for a visual task, and represents non-sensory, task-related signals, (ii) PPC: represents both visual stimulus and response signals, is not required during delay or response period (see concerns about putative requirement of PPC in stimulus period below), (iii) fMC: represents motor planning signals, is required during the delay and response period.

I have several major concerns about interpretation of the data which the authors consider to be novel:

1) The authors argue that PPC multiplexes the stimulus info and behavioural choice, but activity in V1 looks relatively similar, and they do not record from other secondary visual areas, which might contain similar or stronger 'multiplexing'. Therefore it is not clear why PPC is special in this respect. For example, Figure 4 (right) and Figure 5 (red curves) for V1 and PPC do not lot look very different. Thus, if there is any delay period activity in PPC, it is about as much as in V1.

Also, the authors should also discuss the drop in selectivity for the stimulus representation from V1 to PPC.

2) The large fraction of cells are suppressed by the stimulus. What total fraction of cells shows this effect? The extracellular recording control measured only 20 cells, which is quite low. The authors could consider inspecting imaging planes from sites that showed a larger fraction of cells with suppression to ensure these regions are not unhealthy.

3) The claim that the authors have found a requirement for PPC in the stimulus period of the task may be re-considered in the light of the possibility that PPC inhibition may quite likely be inhibiting secondary visual cortices (see discussion in Erlich et al. 2015 on this).

Further, the unusual result of fMC requirement during the stimulus period may be explained by considering what the experimenter refers to as the stimulus period, however, could also be considered as a stimulus plus delay period; considering that mice typically make their choice in 300-600 ms, the remaining period of the 2.2 seconds of silencing will then be a part of the delay period (for the mouse) even though the stimulus may still be on. In effect, this could also be thought of as delay-period silencing.

4) The timing and profile of the behavioral response (licking) of the mice to the stimuli is not transparent. The authors write: 'We applied an a priori exclusion criteria that any mice licking continuously throughout the delay period on target trials would be excluded from the study, since this strategy would possibly obviate the short-term memory component of the task. However, none of the mice used this strategy even in earlier phases of training (see Video 1 for representative mouse performance).' Do mice respond during the stimulus and delay period? Is (attempted) licking measured at all during these two early phases? This is important for interpreting the activity during the delay period (component of movement planning or execution).

Also, did the authors record eye movements? Differences across stimulus conditions in eye movement signals could also contribute to activity differences in visual and motor areas.

Reviewer #2:

In this work Goard and colleagues develop a delayed a very elegant behavioral choice task for mice. They use this task to assess the activity of V1, PPC and fMC during stimulus, delay and response phases of the task. Experiments and analysis seem solid. My main concerns are the interpretations of the data – this is detailed below.

Suggestions for improvement:

1) Subsection ‘Calcium imaging during a memory-guided sensorimotor decision task’. How tropism and promoter of the AAV system influence expression is still a matter of quite some debate (i.e. how many of the cells are excitatory or inhibitory). Unless the authors actually show data, I would remove the speculation that the 'vast majority' of cells were excitatory. Moreover, without quantification the statement is meaningless (with non-selective labeling one would expect roughly 80% of cells to excitatory – and 80% would probably qualify as 'vast majority'). More importantly, the paper does not profit from this assumption.

2) The distinction between stimulus and delay periods is probably not as clear as the authors make it seem. One critical problem of the task design is that the stimulus period is probably much longer than the animal (once trained) would need to make a decision. Most of the sensorimotor transformation is probably already over by the time the stimulus ends (at 2 s) in trained animals. It would probably best to systematically reduce the stimulus period to the minimum duration at which mice still perform above chance.

3) The finding that PPC is necessary during the stimulus period is also confounded, by two things: first, this conclusion is again made difficult by the aforementioned mixing of stimulus and delay phase; second, PPC is directly adjacent to V1 – it is unclear to me how the authors ensure that V1 is not directly affected by the inhibition of PPC.

4) I would rephrase the Introduction, as it makes the unfounded assumption that there is only one region in the brain involved in sensorimotor transformations (by all practical standards, the *only* tasks the brain really needs to solve can be reduced to sensorimotor transformation problems).

Reviewer #3:

The authors simultaneously imaged neural activity (via a fluorescent calcium indicator) from hundreds of neurons in three cortical regions from behaving (head-fixed) mice. To my knowledge, this represents the richest data-set collected from animals performing a delayed-response task. Given the mission of *eLife*, I expect that the data set (*ΔF/F* traces from all trials from all neurons) be made available upon publication. This data set alone is a significant contribution to the field.

The authors imaged visual (V1), parietal (PPC) and frontal (fMC) cortices in a visual perceptual discrimination task. V1 activity was correlated mostly with the stimulus, with good sensory discrimination. Although some neurons in V1 also showed dynamics around the time of the response. In PPC, a substantial number of neurons responded at the time of the stimulus and again at the time of the response. In fMC, few neurons had dynamics at the time of the stimulus, but did respond during the delay and the response.

The population analysis the authors performed revealed a more interesting picture (although they strangely put this in a supplement). Although the PPC neurons contain stimulus and choice information, the information is essentially lost towards the end of the delay period, before the neurons begin to encode choice. In contrast, the fMC neurons continually increase their information about the choice throughout the trial.

To test the causal contribution of the activity in the three regions, the authors optogenetically inactivated each of the regions bilaterally during the three task epochs: stimulus, delay and choice. Disrupting the activity in any of the three regions during the stimulus disrupted the behavioral performance. However, only fMC inactivations were effective during the delay and response period. In fact, silencing of the fMC during the response seemed to completely suppress licking, suggesting that part of the inactivated area may be a motor cortical area for licking.

This manuscript contributes several novel important results which the authors describe in the discussion. One that they did not mention is that this provide new evidence in the controversy over ramps vs. sequences for short-term memory. In Harvey et al. (2012) the PPC neurons seemed to bridge the delay period with a sequence, each neuron firing in turn. This was distinct from the ramps/persistent activity observed in the PPC of primate memory-guided tasks, suggesting a potentially important species difference in working memory mechanisms. The current study, using a visual discrimination task in a head-fixed prep found persistent activity/ramps rather than sequences in PPC, suggesting the sequences observed in Harvey et al., were related to the fact that animals were running through a virtual environment.

I was disappointed that the authors did not do more analyses that took advantage of the large number of simultaneously recorded neurons and the knowledge of the relative physical location of the neurons. However, I imagine that these analyses will be done in a future publication.

In any case, I strongly recommend this manuscript for publication in *eLife*.

Points to address:

1) Population encoding of task-related variables. The authors should generate a proper null hypothesis by performing their analysis with the trials labels permuted several thousand times.

2) I recommend that Figure 2—figure supplement 2 and Figure 5—figure supplement 1 should be main figures.

3) The authors also know the relative physical location of all the neurons with respect to one-another. Is there any spatial organization of the neurons in different clusters?

4) A basic explanation of what the selectivity index is in the text would be appropriate. What does 0 mean? What does 1 mean? It's only one sentence, so you shouldn't force the reader to search through the methods for this.

5) Within the subsection ‘Suppressed neurons exhibit non-selective responses’, the statement 'weakly correlated to the enhanced […] in the region' seems unsupported. Is this referring to the delay modulation index? "In the region" is vague. Given that you know the relative spatial location of the all recorded neurons couldn't you do a very precise estimate of the expected signal from a PV neuron that receives input from nearby neurons?

Reviewer #4:

*eLife’s* criteria state: '…an article in *eLife* might offer fundamental biological insight, brilliant methodological inventiveness, or profound societal benefits.'

Overall: Given *eLife*'s criteria, then this article is not currently suitable for publication. That said, the data are lovely (albeit not 'brilliantly inventive') and an improved discussion/interpretation could warrant publication as a community resource (albeit without 'profound societal benefits').

Data: Recordings from multiple-cortical areas during behaviour are rare and will be absolutely necessary to advance our understanding of cortical function. The functional imaging experiments reported here are an excellent step towards this goal (the data in Figure 4 are very satisfying). However, the behaviour task used here is insufficient to support the authors' conclusions and interpret the photo-inhibition studies.

Task: The authors refer to their behavioural assay as a 'a memory-guided visual discrimination task.' However, they acknowledge in the Discussion that it is not 'a task that requires memory of the stimulus', such as delayed match-to-sample, thus implying that it requires memory of the correct response. This is an incorrect assumption. Permit me the following example:

You are seated at a table upon which there is a computer monitor and a big red button. A scientist in the room explains that 'a command will briefly be displayed on the monitor, if you see the word PRESS, wait until the button lights up, then press it. If you see the words DON'T PRESS, then don't do anything when the button lights up.' The time between the command appearing on the screen and the button illuminating is variable. You can move around however you like. You will be doing this task for 117 days. How do you go about it?

Memory-guided (stimulus): You could keep in mind the written command, remembering whether PRESS or DON'T PRESS was presented, and then decide upon and execute the correct action (press or nothing) when the button illuminates.

Memory-guided (response): You could keep in mind the appropriate response, remembering whether to press or not, and then execute that action when the button illuminates.

Behavioural-strategy: You respond immediately to the presented command. If PRESS is presented, then you put your hand on top of the button. When the button illuminates, if your hand is on it, then you press it and remove your hand. The decision to press is made when the command is given and your response begins immediately. The button lighting-up is simply the cue to carry out the final portion of the action. No memory of the command presented or response selected is required during the delay period.

What do your mice do? Well, we don't know. They could use any of these strategies (or some combination of them), but we just can't tell with your assay. This makes it very difficult to interpret what is represented by neural activity or what is perturbed by photo-inhibition. It would be interesting to monitor their behaviour in much more detail, even to the level of EMG, during the delay period. I would predict that there exists a clear behavioural signature of the future action the animal will perform when presented with the cue (water spout), and that this is disrupted by optogenetic inhibition of fMC.

Interestingly, if a 'behavioural strategy' is used, then it would make sense that sensory, parietal, and motor areas would all be required during the 'command' period; the sensory discrimination, decision, and the action all happen immediately. The action, however, persists (i.e. resting your hand atop the button) and can thus also be perturbed at later stages by disrupting activity in motor areas (Figure 7).

Major comments:

a) Discuss and highlight the caveats of your behavioural paradigm.

b) New experiments: Are there behavioural measures that predict the future response? Are they affected by photo-inhibition?

Note: negative results are difficult to interpret (there could simply be a behavioural measure that you did not consider). However, positive results would be informative and help interpret both the functional imaging and inactivation experiments.

c) The observation that each area appears to (re)activate during the response phase is very interesting and warrants much more analysis/discussion. For example, how do V1 neurons with response phase activity differ from their counterparts with sensory phase activity? Are they more anterior? Are they found at different depths? Similar analysis within and across the V1, PPC, and fMC fields-of-view could provide a unique micro (local) and macro (global) characterization of functional organization possible with your imaging/behaviour setup.

[Editors’ note: what now follows is the decision letter after the authors submitted for further consideration.]

Thank you for submitting your article "Distinct Roles of Visual, Parietal, and Frontal Motor Cortices in Memory-guided Sensorimotor Decisions" for consideration by *eLife*. The Reviewing Editor has considered your appeal with the three reviewers (overseen by the Senior Editor, Tim Behrens).

The following individuals involved in review of your submission have agreed to reveal their identity: Jeffrey Erlich (reviewer 3) and Adam Kampf (reviewer 4).

You will see from the reviews below that there are still major issues, but there is still support. We don't want you to go to the trouble of major new experiments without any guarantee of acceptance. Could you therefore give us a point-by-point rebuttal to the original concerns giving us a plan for how you would also address the reviews below with a particular focus on reviewer 3. This is a particularly difficult case (because we all agree the data are extraordinary and we would all like to be able to publish them, but we are still not convinced that there are not interpretational problems with the experimental design).

Reviewer #2:

I think the only argument that the authors have made that would merit re-evaluation is that they have additional data using shorter stimulus periods.

Reviewer #3:

I am sympathetic to the authors appeal. There are important differences between their memory-guided task, which is a go/no-go 2-stimulus task, and some those used historically. For example, in our study of auditory memory-guided orienting we used multiple stimuli to measure a psychometric curve (Erlich et al., 2011). In many classic monkey studies animals has to remember one of 8 locations. The lack of psychometrics in this study makes it harder to interpret the effects of inactivation, but I generally support the position that this is something they should address in their discussion, rather than be a cause for rejection.

*Reviewer #4:*

Just to be clear, I would be quite happy to work with the authors towards a manuscript appropriate for publication in *eLife*. As stated in my original review, I applaud their multi-cortical area imaging and inactivation dataset, yet I strongly believe that the caveats of their behavioural task must be directly addressed in the manuscript. This will require new analysis (and possibly further experiments) as well as a substantially revised discussion. I will try to be very clear in the following about what I would require, such that the authors' can decide how they wish to proceed.

Regarding the central criticism of the manuscript, i.e. the difficulty in determining the strategy animals use to solve their delayed-response task (behaviour or memory):

The authors argue that if their animals were using an overt "behavioural strategy" (i.e. licking continuously after the GO stimulus), then this would obscure their proposed interpretation. Whereas, if the strategy was less obvious (i.e. a "micro-postural adjustment"), then they are justified in interpreting delay-period activity as memory-related.

Authors: "Researchers studying delayed-responses are well aware that these micropostural adjustments can take place prior to motor action. However, these "premotor adjustments" do not automatically select for an action (the way that resting a hand on the button would), they presumably require cortical activity to maintain and translate to the correct motor action at the correct time, and they in no way eliminate the need to remember the correct response."

I quite disagree. After over a hundred days of training and with only two actions to select from (lick or not-lick), then it is easy to imagine a mouse devising a simple "trick" for solving your task that its entirely independent of cortically-maintained short-term memory. Consider the following scenario:

"When the GO stimulus is presented, a mouse responds by contracting its tongue muscles to lower its already low threshold for licking (Figure 1, ~40% false alarms). The presentation of the spout simply triggers a sensitized lick generator. Cortical activity is only necessary to lower the threshold for licking during GO stimulus presentation and is not maintaining a short-term memory of the selected response."

It is easy to imagine other strategies, some more conspicuous than others: the mouse moves its whiskers after the GO stimulus to intercept the spout, directs its eyes to the visual field where it is most sensitive to spout motion, etc. The possibility that some subtle "behavioural strategy" is being used by these mice must be explicitly addressed! As the author's note, researchers using delayed-response tasks are well aware of this dilemma. However, and this is unclear to me, they seem to imply that because these concerns apply to all previous studies, it is somehow less of a concern (?!).

Authors: "We would also like to point out that this argument could be applied to every delayed-response paper currently published, ranging from classic work on delayed reaching and saccades in nonhuman primates (3,4), to more recent work on delayed-response in rodents (5-7), as no researcher has yet undertaken the (likely intractable) task of ruling out every possible premotor adjustment."

Exactly. To borrow from the 1982 Goldman-Rakic paper, discussing possible interpretations for 'expectancy waves' observed during delay periods (my emphasis):

"On the other hand we cannot rule out that the monkey adopts a covert position or bodily orientation which is held for variable periods depending upon the duration of the delay or may exercise an inhibition of premature, irrelevant or incorrect responses also in relation to the length of the delay. *Only future experimentation in which behavior of the monkey is controlled in different ways can resolve these various possibilities*."

The authors are well aware of these complications, so where is the detailed consideration of possible covert "behavioural strategies" in the manuscript? It is true that in many studies, old and recent, such considerations are often omitted. However, as the authors acknowledge in the Introduction, there remains a substantial amount of confusion in this field:

Authors: "There are several possibilities for the discrepancies seen in the previous studies, including differences in species, sensory modality, and behavioral task."

Rather than add to the confusion, I must insist that you offer some serious attempts at clarification.

What to do: New analysis/experiments

I presume you have a video record of all behaviour (at least during the imaging and inactivation experiments). If not, then new animals will have to be trained.

a) Is there a discernible distinction between animal movement or posture during delay periods preceding a GO vs NO GO response? A "cartoon" example of static posture analysis can be done on your supplementary video by simply averaging all frames during the delay period following GO (green) or NO-GO (red) stimuli and overlaying them.

b) Are changes in whisking, posture, ear orientation, eye gaze (if available), etc. predictive of the animal's eventual response? If so, then how are these "behavioural" changes influenced by photo-inactivation? How do they develop over learning, etc.?

What to discuss: New points to include in your manuscript

Why did you use a delayed-response task versus a delayed-comparison task?

What tasks did you attempt prior to converging on this behaviour paradigm? What influenced your decision?

Why did you not include at least two different stimuli for each response (45 and -45 degree gratings for GO, vertical and horizontal for NO-GO). This would have at least allowed you to distinguish whether cortical delay activity was stimulus specific or related to the future response.

Why did you not use a two alternative choice task with more than one active response (licking left vs. licking right) as in Guo et al. 2013?

And, of course, the results of your analysis of the behaviour.

[Editors’ note: what now follows is the decision letter after the authors submitted for further consideration.]

Thank you for choosing to send your work entitled "Distinct Roles of Visual, Parietal, and Frontal Motor Cortices in Memory-guided Sensorimotor Decisions" for consideration at *eLife*.

The Reviewing Editor has reconsidered your article and discussed it with the Senior Editor (Timothy Behrens).

We appreciate the effort you have gone to propose changes to the manuscript in a revision. The changes you have proposed seem appropriate, but not complete. Here is the situation as we see it.

1) The argument that says "Many other people have done similar things in the past" is not an argument that we or the reviewers find persuasive. We understand that there are similarities, but the reviewers have also highlighted differences. More to the point, we do not see it as a relevant point. The relevant point is whether the claims can be supported.

2) Often at *eLife* we are in the position where one (or even at a push two) reviewer(s) takes a strong position on a particular point, but due to the strength of the *eLife* discussion process the reviewers can come to an arrangement where this is highlighted in the discussion but does not require extra data or analyses. This is really not the case here. The editors want to emphasize that three out of four reviewers have highlighted similar issues. Furthermore, during the triage process, two of the BREs consulted, despite supporting the manuscript broadly, raised similar potential issues. In total four people out of six who we have consulted on this manuscript have held similar views. However strongly you argue your case, it is not within our power to overturn this much weight of opinion on the basis of argument alone.

3) The decision we have reached is therefore that we cannot continue with the manuscript unless you address the principal issue as suggested by reviewer 4. That is: There is no point in resubmitting the paper unless the analysis of the videos of behavior shows no postural and lick differences in the delay period.

If you do this analysis, we would be happy to re-consult with the referees.

---

## [Author Response]

[Editors’ note: this article was originally rejected after discussions between the reviewers, but the authors were invited to resubmit after an appeal against the decision. What follows is the letter of rebuttal.]

Thank you for taking the time to conduct a thorough review of our submitted manuscript. Although we were very impressed by the quality and content of the reviewer comments, we were disappointed to find that the central criticism, and ultimately the reason given for rejecting the manuscript, was based on a questionable argument about the validity of our behavioral task (as laid out by reviewer #4). We believe we can concisely show the that the reviewer's argument is erroneous, and will revise the paper accordingly so that readers do not fall prey to the same conclusion (please see our response to reviewer #4’s concerns below). It should be noted that, if taken at face value, the reviewer's argument would not only apply to our task but to every delayed-response task used in the last five decades of neuroscience research, ranging from classics in the field (Fuster & Alexander, Science, 1971; numerous papers from Goldman-Rakic and colleagues) to recent high-impact articles from the Svoboda and Tank labs (e.g., Harvey et al., Nature, 2012; Guo et al., Neuron, 2013; Li et al., Nature, 2015).

In addition, to respond to the central criticism, we can provide new data that directly addresses many of the more minor issues that the reviewers raised. In particular, by presenting data in which we used a very short (250ms) stimulus, we can directly address the concern that the stimulus was too long and was therefore conflated with the delay period.

Response to reviewer #4:

We would like to thank the reviewer for concisely summarizing the central criticism to our behavioral task. This was very helpful in understanding the concern, and will help us clarify why it is not an accurate comparison. To summarize the reviewer’s argument:

“You are seated at a table upon which there is a computer monitor and a big red button. A scientist in the room explains that "a command will briefly be displayed on the monitor, if you see the word PRESS, wait until the button lights up, then press it. If you see the words DON'T PRESS, then don't do anything when the button lights up." The time between the command appearing on the screen and the button illuminating is variable. You can move around however you like. You will be doing this task for 117 days. How do you go about it?

Memory-guided (stimulus): You could keep in mind the written command, remembering whether PRESS or DON'T PRESS was presented, and then decide upon and execute the correct action (press or nothing) when the button illuminates.

Memory-guided (response): You could keep in mind the appropriate response, remembering whether to press or not, and then execute that action when the button illuminates.

Behavioural-strategy: You respond immediately to the presented command. If PRESS is presented, then you put your hand on top of the button. When the button illuminates, if your hand is on it, then you press it and remove your hand. The decision to press is made when the command is given and your response begins immediately. The button lighting-up is simply the cue to carry out the final portion of the action. No memory of the command presented or response selected is required during the delay period.”

We agree that if the mice were employing the “behavioral strategy”, it would certainly obscure the interpretation of the findings and reduce the relevance to the understanding of delayed- response behaviors. This strategy would not require any sort of memory during the delay period, and likely any cortical activity during the delay period would be entirely superfluous (as the correct motor action has already been made).

However, we must point out that the mice are most certainly not using this strategy as stated by the reviewer. The mice could perform the task this way, by sticking out their tongue (or possibly by continuously licking) from the time the stimulus is presented until the time the spout comes forward. However, as was made clear in the manuscript, the mice do not perform the task in this manner. Instead, they wait for the spout to come forward and only then commence licking.

We believe the reviewer already appreciated this point, and was instead making a more subtle argument, specifically, that the mice are making some micropostural adjustment (e.g., changing the position of their jaw or tongue) that is serving as a proxy for the eventual motor action.

However, this is a very different situation, and comparing it to resting a hand on the response button is a false equivalency. Researchers studying delayed-responses are well aware that these micropostural adjustments can take place prior to motor action. However, these “premotor adjustments” do not automatically select for an action (the way that resting a hand on the button would), they presumably require cortical activity to maintain and translate to the correct motor action at the correct time, and they in no way eliminate the need to remember the correct response. Indeed, extensive work from the labs of Krishna Shenoy, Mark Churchland, and others (1,2) have shown that this preparatory activity is not simply a reduced version of the eventual motor activity, but involves the firing of a very different subset of neurons, a result recapitulated in our manuscript (Figure 2, I-L; Figure 4, bottom).

We would also like to point out that this argument could be applied to every delayed-response paper currently published, ranging from classic work on delayed reaching and saccades in non- human primates (3,4), to more recent work on delayed-response in rodents (5-7), as no researcher has yet undertaken the (likely intractable) task of ruling out every possible premotor adjustment. For example, consider the recent paper from the Svoboda lab (7) in which a somatosensory stimulus cues the mouse to lick left or right after a fixed delay period. In this case, there is no reason the mice cannot make a premotor adjustment (e.g., positioning their tongue on the left or right side of their mouth), and yet again the authors found that distinct populations of neurons are active during the premotor and motor periods of the task, and that both sets of neurons play distinct roles in task performance, consistent with a necessary role for motor planning prior to execution.

In summary, we very much appreciate the reviewers’ discussion of this complex issue, and agree that it deserves additional clarification in the Discussion section of our paper. In particular, we will address the possibility of preparatory adjustments, and the role they may play in motor planning in our study and others. However, we strongly believe that rejecting our manuscript on the basis of this criticism (one that applies equally to the last 40+ years of research on delayed responses) is not appropriate, and would urge the reviewers and editors to reconsider.

References:

1. Churchland, et al. Cortical preparatory activity: representation of movement or first cog in a dynamical machine? Neuron, 2010.

2. Kaufman, et al. Cortical activity in the null space: permitting preparation without movement. Nature Neurosci, 2014.

3. Fuster & Alexander, Neuron activity related to short-term memory. Science, 1971.

4. Kojima & Goldman-Rakic, Delay-related activity of prefrontal neurons in rhesus monkeys performing delayed response. Brain Res, 1982.

5. Harvey et al. Choice-specific sequences in parietal cortex during a virtual-navigation decision task. Nature, 2012.

6. Guo et al. Flow of cortical activity underlying a tactile decision in mice. Neuron, 2013.

7. Li et al. A motor cortex circuit for motor planning and movement. Nature, 2015.

[Editors’ note: the author responses to the first and second rounds of peer review follow.]

We would like to thank all four reviewers for their time and helpful comments. We hope that the changes we have made and the new data we have added to the manuscript will alleviate any major concerns. In the interest of accelerating the review process, the more time-intensive revision requests will be made as requested in the next version of the manuscript (these are indicated in the response letter below). Revisions that have already been completed are highlighted in the uploaded manuscript and indicated below.

Reviewer #1:

*In order to understand the role of three distinct cortical regions in a sensorimotor transformation, Goard et al. recorded from V1, PPC and fMC of mice performing a visual discrimination task with a response delay introduced by removing the lick spout for varying times.*

*A number of findings from this study are re-establishing known findings. This includes findings about (i) V1: visually responsive and required for a visual task, and represents non-sensory, task-related signals, (ii) PPC: represents both visual stimulus and response signals, is not required during delay or response period (see concerns about putative requirement of PPC in stimulus period below), (iii) fMC: represents motor planning signals, is required during the delay and response period.*

*I have several major concerns about interpretation of the data which the authors consider to be novel:*

*1) The authors argue that PPC multiplexes the stimulus info and behavioural choice, but activity in V1 looks relatively similar, and they do not record from other secondary visual areas, which might contain similar or stronger 'multiplexing'. Therefore it is not clear why PPC is special in this respect. For example, Figure 4 (right) and Figure 5 (red curves) for V1 and PPC do not lot look very different. Thus, if there is any delay period activity in PPC, it is about as much as in V1.*

Also, the authors should also discuss the drop in selectivity for the stimulus representation from V1 to PPC.

We agree that PPC does not show dramatically more delay period activity than V1 (11% vs 6% of neurons, Figure 2—figure supplement 2). However, this is independent of the encoding of behavioral choice, which can occur in neurons responding during the stimulus period (see Figure 5 during stimulus epoch). Our lab has further confirmed the difference in stimulus/choice encoding of V1 and PPC neurons in a zero-delay version of the task as part of a second study under review (Pho et al., in review). For example, we found that if the target and non-target stimuli are switched, many of the neurons in PPC (but not V1) switch their response preferences (Pho et al., Figure 7). We will include this information in our discussion of the multiplexing within PPC (seventh paragraph of the Discussion).

The reason we see a drop in stimulus representation in PPC is likely because many of the stimulus coding neurons also show modulation by other non-sensory factors such as engagement (Pho et al., Figure 6) and choice (Figure 5; Pho et al., Figure 7). We will add this to our text to enhance the Discussion.

2) The large fraction of cells are suppressed by the stimulus. What total fraction of cells shows this effect? The extracellular recording control measured only 20 cells, which is quite low. The authors could consider inspecting imaging planes from sites that showed a larger fraction of cells with suppression to ensure these regions are not unhealthy.

The total fraction of cells showing suppression was 975/3049 neurons (32.0%). We have added this information to the text (Results section, subsection “Suppressed neurons exhibit non-selective responses”). Manual inspection of neurons exhibiting suppressed activity showed healthy expression of GCaMP6s (nucleus-excluded) and normal decay times for fluorescence transients. If the reviewer thinks it would be helpful, we would be happy to add this data to Figure 2—figure supplement 1 in the subsequent submission.

*3) The claim that the authors have found a requirement for PPC in the stimulus period of the task may be re-considered in the light of the possibility that PPC inhibition may quite likely be inhibiting secondary visual cortices (see Discussion in Erlich et al. 2015 on this).*

Further, the unusual result of fMC requirement during the stimulus period may be explained by considering what the experimenter refers to as the stimulus period, however, could also be considered as a stimulus plus delay period; considering that mice typically make their choice in 300-600ms, the remaining period of the 2.2 seconds of silencing will then be a part of the delay period (for the mouse) even though the stimulus may still be on. In effect, this could also be thought of as delay-period silencing.

Although the photoinhibition approach is probably the best method for testing the necessity of a specific cortical region, we agree that the results must be treated with some caution, and have added text to improve the Discussion.

In order to address the concern about the combined stimulus/delay period, we have carried out additional experiments in which we trained mice to perform the same task with a 250 ms stimulus (stimuli shorter than 250 ms resulted in compromised behavioral performance). Similar to the experiments with longer stimulus times, we found that photoinhibition during this brief window perturbed performance for all three cortical areas (Figure 9—figure supplement 1). However, please note that recovery of activity after photoinhibition can take several hundred milliseconds (τ = 416 +/- 91 ms), so the results should be interpreted with caution. However, these data do demonstrate that fMC is necessary either during or very shortly after the visual stimulus is shown, highlighting the rapid conversion of sensory into motor information across cortical regions.

*4) The timing and profile of the behavioral response (licking) of the mice to the stimuli is not transparent. The authors write: 'We applied an a priori exclusion criteria that any mice licking continuously throughout the delay period on target trials would be excluded from the study, since this strategy would possibly obviate the short-term memory component of the task. However, none of the mice used this strategy even in earlier phases of training (see Video 1 for representative mouse performance).' Do mice respond during the stimulus and delay period? Is (attempted) licking measured at all during these two early phases? This is important for interpreting the activity during the delay period (component of movement planning or execution).*

*Also, did the authors record eye movements? Differences across stimulus conditions in eye movement signals could also contribute to activity differences in visual and motor areas.*

We are currently analyzing the video taken during behavior to look for differences in movements during the delay period (see response to reviewer #4). This analysis will be included in the revised submission as a supplementary figure. We did not record eye movements, but eye movement recordings during a similar task showed no significant modulation by trial type (Andermann, Kerlin & Reid, Front Cell Neurosci, 2010).

Reviewer #2:

*In this work Goard and colleagues develop a delayed a very elegant behavioral choice task for mice. They use this task to assess the activity of V1, PPC and fMC during stimulus, delay and response phases of the task. Experiments and analysis seem solid. My main concerns are the interpretations of the data – this is detailed below.*

*Suggestions for improvement:*

1) Subsection “Calcium imaging during a memory-guided sensorimotor decision task”. How tropism and promoter of the AAV system influence expression is still a matter of quite some debate (i.e. how many of the cells are excitatory or inhibitory). Unless the authors actually show data, I would remove the speculation that the "vast majority" of cells were excitatory. Moreover, without quantification the statement is meaningless (with non-selective labeling one would expect roughly 80% of cells to excitatory – and 80% would probably qualify as "vast majority"). More importantly, the paper does not profit from this assumption.

During presentations of this work at conferences, we often received questions about whether the different types of responses are directly related to neuronal cell type, so we believe it is important to clarify that this appears not to be the case. In the subsequent submission, we will include a supplementary figure showing the weak responses of PV+ and SOM+ neurons measured with volume scanning. If the supplementary figure is not sufficiently convincing, then we will remove the statement, but at this point we believe it does add value and would prefer to keep it.

2) The distinction between stimulus and delay periods is probably not as clear as the authors make it seem. One critical problem of the task design is that the stimulus period is probably much longer than the animal (once trained) would need to make a decision. Most of the sensorimotor transformation is probably already over by the time the stimulus ends (at 2s) in trained animals. It would probably best to systematically reduce the stimulus period to the minimum duration at which mice still perform above chance.

We agree with this criticism, and have included new data to address this issue. We trained animals to perform the discrimination with a 250 ms stimulus (shorter durations led to very poor behavioral performance). We then tested the effect of inactivating regions V1, PPC, and fMC during the shortened stimulus. We found a striking reduction in behavioral performance in all three areas, similar to the results using longer stimuli (Figure 9—figure supplement 1). One caveat is that recovery from photoinhibition can take several hundred milliseconds (τ = 416 +/- 91 ms), and is variable between neurons, so there may be a suppressive effect outlasting the stimulus.

However, these data do confirm that fMC is involved very early in the trial.

3) The finding that PPC is necessary during the stimulus period is also confounded, by two things: first, this conclusion is again made difficult by the aforementioned mixing of stimulus and delay phase; second, PPC is directly adjacent to V1 – it is unclear to me how the authors ensure that V1 is not directly affected by the inhibition of PPC.

The first issue is hopefully resolved by the new data. As far as the second issue, the retinotopically matched region of V1 is 1.5-2 mm away from PPC and is unlikely to be directly affected by the photoinhibition. However, we cannot rule out polysynaptic effects, particularly in light of recent experiments from the Olvecksy lab (Otchy et al. 2015). We have added discussion of these concerns to the Discussion (eighth paragraph of Discussion).

*4) I would rephrase the Introduction, as it makes the unfounded assumption that there is only one region in the brain involved in sensorimotor transformations (by all practical standards, the only tasks the brain really needs to solve can be reduced to sensorimotor transformation problems).*

We fully agree, and did not intend to make this assumption. We could not find specifically where it was stated in the Introduction, but we have changed the wording in order to avoid giving that impression. Please let us know if you have additional suggestions.

Reviewer #3:

*The authors simultaneously imaged neural activity (via a fluorescent calcium indicator) from hundreds of neurons in three cortical regions from behaving (head-fixed) mice. To my knowledge, this represents the richest data-set collected from animals performing a delayed-response task. Given the mission of eLife, I expect that the data set (ΔF/F traces from all trials from all neurons) be made available upon publication. This data set alone is a significant contribution to the field.*

*The authors imaged visual (V1), parietal (PPC) and frontal (fMC) cortices in a visual perceptual discrimination task. V1 activity was correlated mostly with the stimulus, with good sensory discrimination. Although some neurons in V1 also showed dynamics around the time of the response. in PPC, a substantial number of neurons responded at the time of the stimulus and again at the time of the response. In fMC, few neurons had dynamics at the time of the stimulus, but did respond during the delay and the response.*

*The population analysis the authors performed revealed a more interesting picture (although they strangely put this in a supplement). Although the PPC neurons contain stimulus and choice information, the information is essentially lost towards the end of the delay period, before the neurons begin to encode choice. In contrast, the fMC neurons continually increase their information about the choice throughout the trial.*

*To test the causal contribution of the activity in the three regions, the authors optogenetically inactivated each of the regions bilaterally during the three task epochs: stimulus, delay and choice. Disrupting the activity in any of the three regions during the stimulus disrupted the behavioral performance. However, only fMC inactivations were effective during the delay and response period. In fact, silencing of the fMC during the response seemed to completely suppress licking, suggesting that part of the inactivated area may be a motor cortical area for licking.*

*This manuscript contributes several novel important results that the authors describe in the Discussion. One that they did not mention is that this provides new evidence in the controversy over ramps vs. sequences for short-term memory. In Harvey et al. (2012) the PPC neurons seemed to bridge the delay period with a sequence, each neuron firing in turn. This was distinct from the ramps/persistent activity observed in the PPC of primate memory-guided tasks, suggesting a potentially important species difference in working memory mechanisms. The current study, using a visual discrimination task in a head-fixed prep found persistent activity/ramps rather than sequences in PPC, suggesting the sequences observed in Harvey et al., were related to the fact that animals were running through a virtual environment.*

*I was disappointed that the authors did not do more analyses that took advantage of the large number of simultaneously recorded neurons and the knowledge of the relative physical location of the neurons. However, I imagine that these analyses will be done in a future publication.*

In any case, I strongly recommend this manuscript for publication in eLife.

We thank the reviewer for their comments. We would be happy to make all data available in a Dryad digital repository following publication. In accordance with the comments of the reviewer, we have added discussion of sequential versus persistent activity to the text (Third paragraph of Discussion). Other comments brought up in the Introduction are addressed below.

*Points to address:*

1) Population encoding of task-related variables. The authors should generate a proper null hypothesis by performing their analysis with the trials labels permuted several thousand times.

Thank you for the suggestion, we will replace our current significance testing method with the suggested analysis in the subsequent submission.

2) I recommend that Figure 2—figure supplement 2 and Figure 5—figure supplement 1 should be main figures.

We would be happy to move these figures to the main text in the subsequent submission.

3) The authors also know the relative physical location of all the neurons with respect to one-another. Is there any spatial organization of the neurons in different clusters?

We include some analysis of spatial location within PPC in a second manuscript from our lab (Pho et al.), but did not find much spatial clustering. This was generally true for V1 and fMC as well. With newly available GCaMP6 transgenic mice, it should be possible to comprehensively map response types to stereotaxic location, but these experiments beyond the scope of the current study.

4) A basic explanation of what the selectivity index is in the text would be appropriate. What does 0 mean? What does 1 mean? It's only one sentence, so you shouldn't force the reader to search through the methods for this.

We have made the correction.

*5) Within the subsection “Suppressed neurons exhibit non-selective responses”, the statement "weakly correlated to the enhanced […] in the region" seems unsupported. Is this referring to the delay modulation index? "In the region" is vague. Given that you know the relative spatial location of the all recorded neurons couldn't you do a very precise estimate of the expected signal from a PV neuron that receives input from nearby neurons?*

If the suppression is from increased inhibition (resulting from increased local activity), we would expect the enhanced and suppressed responses to have similar time courses. Instead we find that the latency for suppressed neurons is very different from the latency for enhanced neurons (e.g., ~400ms vs ~100ms in V1, ~700ms vs ~200ms in PPC), and that the suppressed responses are much more delay-sensitive than the enhanced responses. We have changed the text to make this point more clearly.

Reviewer #4:

*[…] What to do: New analysis/experiments*

*I presume you have a video record of all behaviour (at least during the imaging and inactivation experiments). If not, then new animals will have to be trained.*

a) Is there a discernible distinction between animal movement or posture during delay periods preceding a GO vs NO GO response? (A "cartoon" example of static posture analysis can be done on your supplementary video by simply averaging all frames during the delay period following GO (green) or NO-GO (red) stimuli and overlaying them).

b) Are changes in whisking, posture, ear orientation, eye gaze (if available), etc. predictive of the animal's eventual response? If so, then how are these "behavioural" changes influenced by photo-inactivation? How do they develop over learning? etc.

We thank the reviewer for suggesting this analytic approach. This analysis will require a bit of time to carry out correctly, but we believe it will improve the manuscript and are happy to include it in the subsequent submission.

We went through our video data, and although we do not have video for every behavior experiment (this would have been thousands of hours of video), we do have videos for a number of sessions across multiple mice for both standard and inactivation sessions. We expect this data will be more than sufficient to detect postural changes during the delay period, and will include the results in the subsequent submission.

What to discuss: New points to include in your manuscript

Discuss and highlight the caveats of your behavioural paradigm.

We have added a discussion of the caveats of our behavioral paradigm, including a summary of many of the issues discussed during the review process (Discussion section). We hope an open discussion will prompt further investigation into this unresolved issue with delayed-response tasks.

Why did you use a delayed-response task versus a delayed-comparison task?

This is more a choice of research question (short-term memory for motor vs. sensory information) rather than a choice of technique. We believe that both types of tasks are important to the study of complex behavior – and have added this point to the Discussion.

What tasks did you attempt prior to converging on this behaviour paradigm? What influenced your decision?

Our strategy was to build on an existing visual discrimination task (Andermann et al., 2010) to allow for the study of memory-guided responses by adding a delay period. Although we describe this in part in the Introduction, we are happy to enhance the text based on your recommendations if the current version is insufficient.

Why did you not include at least two different stimuli for each response (45 and -45 degree gratings for GO, vertical and horizontal for NO-GO). This would have at least allowed you to distinguish whether cortical delay activity was stimulus specific or related to the future response.

We did a related experiment in a second study from our lab, using varying contrasts rather than multiple orientations, with the rationale being that visual responses would exhibit contrast- sensitivity while motor responses would not (see Pho et al., Figure 5,Figure 6 for results).

Why did you not use a two alternative choice task with more than one active response (licking left vs. licking right) as in Guo et al. 2013?

We found it added substantially to the training time without getting around the major drawback of delayed-response tasks (the covert postural adjustments discussed earlier).

[Editors’ note: the author responses to the third round of peer review follow.]

We have added a new Figure 1—figure supplement 1 and the following to the Behavior section of the Materials and methods:

“For analysis of movement during the delay period (Figure 1—figure supplement 1), cropped video frames (300 x 200 pixels; width x height) from Hit and CR trials were compared to a “template” CR image to measure postural changes or increases in movement during each epoch of the task (Prestimulus, Delay, Response). […] Note that since the frames are compared against a CR template, this approach will capture not only transient movement but also stable postural changes specific to Hit trials. To compare between sessions, the subtracted difference maps were averaged across all pixels for each epoch (Figure 1—figure supplement 1).”